# Genomic and physiological signatures of adaptation in pathogenic fungi

Marco Alexandre Guerreiro [1,2] ✉, Andrey Yurkov [3], Minou Nowrousian [4], Kirk Broders [5] & Eva H. Stukenbrock [1,2]

Emerging fungal pathogens have detrimental impacts on crops, animals, and humans. Despite the mounting threat of these emerging fungal pathogens, little is known about their transition from saprotrophic to pathogenic lifestyles. To gain insights into fungal lifestyle transitions, we study the *Trichosporonales* order, which includes both saprotrophic species and opportunistic human pathogens, as a system to reveal evolutionary adaptations leading to virulence in fungi. Here we use comparative genome analyses and experimental assays to demonstrate that the transition from saprotroph to opportunistic human pathogen is facilitated by adaptive translation. Codon optimization of metabolic genes grants these fungi the ability to quickly adapt to new environments. In this study, we link genomic data with fungal physiology, highlighting the role of adaptive translation in colonizing different environments and suggesting that gene translation optimization plays a critical role in fungal lifestyle evolution.

Fungi are an emerging and fast-growing threat to human and plant health across the globe[1,2]. Fungi can adapt rapidly in response to new selective pressures in changing environments, a feature enabled by remarkable genomic and phenotypic plasticity. Similarly, fungal pathogens quickly adapt to host physiology and defenses. In clinical and agriculture scenarios, the capacity to rapidly adapt allows fungi to infect new hosts, develop new infection mechanisms and virulence factors, and increase resistance to inhibitors[3–5].

While some fungal species are highly adapted to human and animal hosts, such as *Candida*, *Malassezia*, or *Trichophyton*, many species are of an environmental origin and are opportunistic pathogens, such as some species of *Cryptococcus*, *Trichosporon*, *Histoplasma*, *Blastomyces*, or *Aspergillus*[1,6–9]. A key feature that defines human pathogens is the ability to survive and disseminate within a given host physiology and body temperature[10]. Some saprotrophic species (i.e., which acquire nutrients from dead organic matter) can become opportunistic human pathogens through evolutionary adaptations that enable colonisation and survival both in human hosts and in various natural environments[11,12]. While the opportunistic human pathogens

*Cryptococcus deneoformans* and *Cryptococcus gattii* (Basidiomycota) are commonly isolated from environmental sources such as trees or soil[13], the ability to grow at temperatures up to 40 °C gives them the ability to colonise human and animal hosts, causing systemic infections. In contrast, the exclusively saprotrophic sister species *Cryptococcus amylolentus* and *Cryptococcus floricola* can only grow at temperatures up to 30 °C[14–16].

Several studies have documented that changes in gene expression patterns are core components of adaptation to new environments[17,18]. These gene expression changes can be demonstrated via comparative transcriptomic analyses or be reflected in the tRNA pool, which rapidly evolves to meet new translational requirements. This level of adaptation involves changes in the relative numbers of the different tRNA molecules encoded by the genome[17]. Analyses of tRNA-mediated processes have provided valuable insights into evolutionary mechanisms of fungal adaptation[19] and metabolic gene-trait associations[20].

Protein synthesis is regulated in part by the availability of tRNAs that match the codons being translated. This regulation is possible due to codon usage biases, in which synonymous codons are used in

[1]Environmental Genomics Group, Botanical Institute, Christian-Albrechts University of Kiel, Kiel, Germany. [2]Max Planck Institute for Evolutionary Biology, Plön, Germany. [3]Leibniz Institute DSMZ-German Collection of Microorganisms and Cell Cultures, Braunschweig, Germany. [4]Department of Molecular and Cellular Botany, Ruhr University Bochum, Bochum, Germany. [5]USDA, Agricultural Research Service, National Center for Agricultural Utilization Research, Mycotoxin Prevention and Applied Microbiology Research Unit, 1815 N. University, Peoria, IL, USA. ✉e-mail: mguerreiro@bot.uni-kiel.de

unequal frequencies throughout the genome[21]. This fine-tuning between the available tRNAs and the codon usage bias allows for the regulation of protein expression from individual genes. Highly expressed genes have an optimal codon composition that corresponds to abundant tRNAs, promoting faster and more efficient translation[22]. Consequently, synonymous mutations in coding regions can be selected to optimise codon usage relative to the available tRNA pool, increasing the translation efficiency of key proteins[23,24]. Codon frequencies and tRNA copy number may thus provide insight into selection signatures of codon optimisation and evolutionary adaptation to new environments[25]. Codon optimisation drives genome adaptation in fungal pathogens and is linked to the ability to colonise multiple hosts[19]. However, little is known about fungal lifestyle dynamics and the evolutionary mechanisms that may allow rapid transitions between lifestyles in some species.

The basidiomycetous order *Trichosporonales* (*Agaricomycotina*, *Tremellomycetes*) comprises many species widely distributed in nature. This order includes several species with clinical, agricultural, and biotechnological importance[26]. Further, *Trichosporonales* are emerging pathogens that can cause superficial skin irritations but also invasive, life-threatening, systemic infections. Despite their importance and potential threat, little is known about the evolution, ecology, mechanisms of virulence, and transition to pathogenic lifestyles of fungi belonging to the *Trichosporonales*. In particular, the genus *Trichosporon* (*Trichosporonales*) contains several known human and animal yeast-like pathogens[26]. Superficial and invasive infections by *Trichosporon* species are emerging life-threatening conditions[6,8,9], with an estimated 77% mortality rate for invasive infections[27]. Despite the epidemiological significance of *Trichosporon* infections, few pathogenicity traits are known. While the ability to form biofilm, broad antimycotic resistance, and growth at high temperatures have been reported for these species[28–30], adaptive traits related to pathogenicity have not been identified in this genus or related genera. Given that pathogenic species are found across different genera, it can be assumed that lifestyle transitions have occurred independently at multiple times. The underlying mechanisms that allow fast lifestyle transitions are not known, but this knowledge is critical to help predict the emergence of new pathogens.

We investigated the evolutionary events and genomic and physiological features driving the transition from saprotrophic to human pathogenic lifestyles in fungi. The primary aim of our study was to identify genomic signatures and physiological traits associated with lifestyle and host/substrate specialisation that could be used to identify potential human fungal pathogens. To this end, we characterised the genomic composition of a set of closely related *Trichosporonales* species and performed experimental assays to identify genetic and physiological traits putatively associated with recent lifestyle transitions. Based on their lifestyle, we hypothesised that genomes of saprotrophic species have more genes related to carbohydrate transport and metabolism, while genomes of opportunistic pathogens have higher diversity and copy numbers of genes associated with lipid transport and metabolism. In addition, we hypothesised that lifestyle adaptations are imprinted in the translation machinery, namely in tRNA pools and codon optimisation in pathways relevant to the lifestyle. Due to the ecological versatility of these species, some plasticity is expected in terms of gene regulation and translation, as well as both independent and convergent evolutionary events defining adaptations of species with similar ecological strategies. Furthermore, we hypothesise that opportunistic pathogens have a higher fitness in lipid-rich substrates than saprotrophic fungi.

In this work, we describe evidence that opportunistic fungal pathogens are not strictly defined by gene repertoires, but rather by their ability to readily evolve and adapt to new environments through adaptive translation. In addition, by performing growth assays, we provide experimental evidence that codon optimisation is reflected in fungal fitness.

## Results

### Opportunistic pathogens emerged independently across genera

To determine the evolutionary relationship among members of the *Trichosporonales*, we used publicly available genomic data for 50 individuals, representing 37 different species. Based on a high content of duplicated single-copy orthologous genes (duplicated BUSCOs ≥43%), five individuals were predicted to be hybrids or diploids[31,32] and therefore removed from further analyses (Supplementary Fig. 1). Notably, among the excluded individuals, we identified a hybrid/diploid genome belonging to the opportunistic pathogen *Cutaneotrichosporon dermatis* (Supplementary Fig. 1).

We conducted a phylogenomic analysis based on 45 haploid genomes (Supplementary Data 1), including 1438 single-copy orthologous protein sequences present in all genomes. The inferred phylogenomic tree topology indicates that the different genera are monophyletic (bootstrap support ≥99%) with opportunistic pathogenic species mainly found in the *Trichosporon* and *Cutaneotrichosporon* genera and 2 additional species found in the *Apiotrichum* genus (Fig. 1). The distribution of pathogenic species across the phylogeny and the ancestral state reconstruction (Supplementary Fig. 2) suggest that the ability to infect human hosts has emerged independently multiple times during the evolution of the *Trichosporonales* order. Furthermore, the genera *Cryptococcus*, *Trichosporon*, and *Cutaneotrichosporon* contain species with clearly defined and experimentally supported lifestyles, either as exclusive saprotrophs (e.g., *Cryptococcus amylolentus*) or as common and well-characterised opportunistic pathogens (e.g., *Cryptococcus deneoformans*)[33]. However, the genus *Apiotrichum* contains mainly saprotrophic species but also includes species which recently were recognised as emerging opportunistic pathogens (i.e., *Apiotrichum mycotoxinivorans* and *Apiotrichum veenhuisii*).

### The genome composition of saprotrophs and pathogens is indistinguishable

To identify genomic signatures associated with lifestyle and with convergent evolution driven by similar ecological environments, we characterised and compared the genomic structure and composition between opportunistic pathogens and saprotrophic species in the order *Trichosporonales*.

The haploid genome assemblies among the *Trichosporonales* species were estimated to range from 17.22 Mb to 33.70 Mb (Figs. 1, 2). We found a significant correlation (Spearman's Rho = 0.62, $P = 0.0067$) between genome size and transposable elements (TEs) in the genome, suggesting that TE content is a determinant of genome expansion (Supplementary Fig. 3).

We next investigated whether the predicted fungal lifestyles (saprotrophic or opportunistic pathogen) were reflected in genome structures or genomic feature content among *Trichosporonales* species. We further annotated the genomes to compare gene composition related to carbohydrate and lipid metabolism. While the genomic features varied considerably between species, we found no significant differences (Wilcoxon test $P ≥ 0.05$) between saprotrophic and opportunistic pathogens (Fig. 2, Supplementary Fig. 4 and Supplementary Data 2). Intriguingly, the number of genes encoding carbohydrate-active enzymes (CAZymes), genes encoding secreted proteins, and genes involved in carbohydrate and lipid transport and metabolism were similar across saprotrophic and pathogenic species (Supplementary Data 2). We compared the genomic composition of these pathways in saprotrophic and opportunistic pathogenic species of *Trichosporonales* and *Tremellales*. While saprotrophic *Cryptococcus* species encoded more secreted proteins and CAZymes compared to

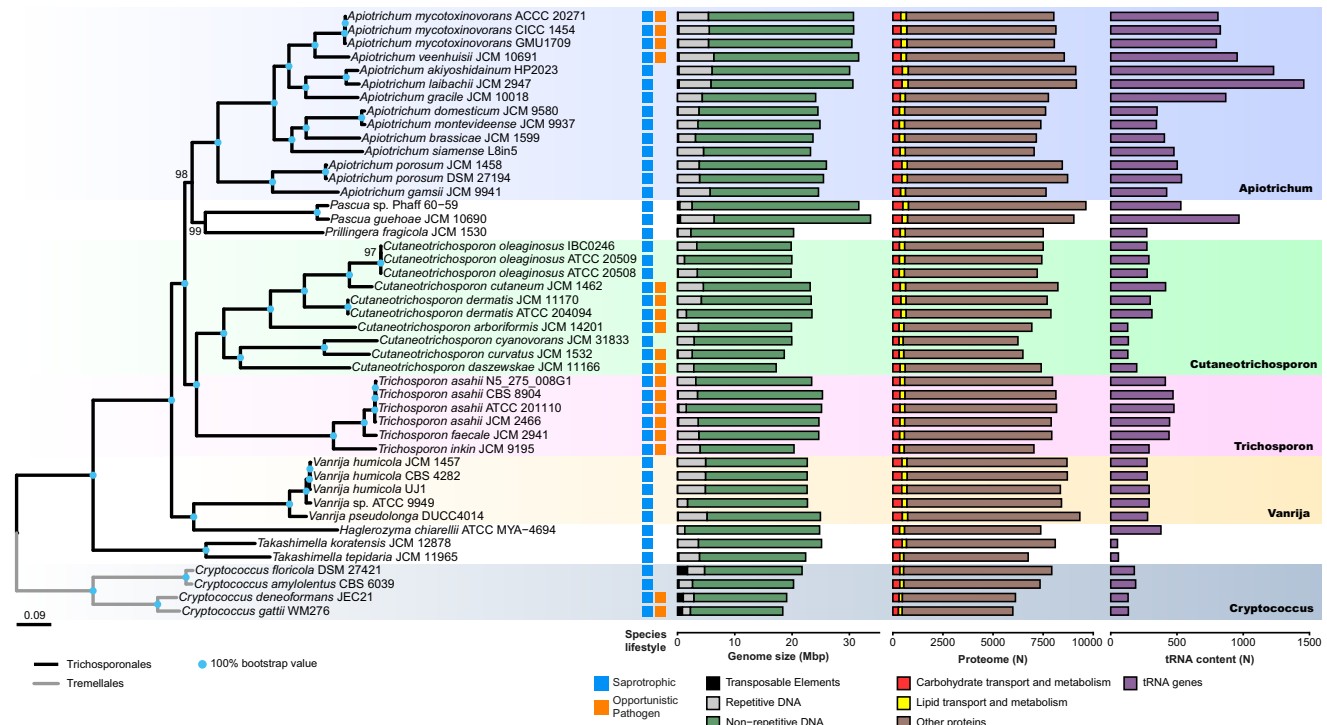

**Fig. 1 | Overview of the analysed haploid *Trichosporonales* species and their respective genomic features.** A maximum likelihood genome-scale phylogeny of the *Trichosporonales* (black branches) and *Tremellales* (grey branches) was calculated based on the protein sequence of 1438 single-copy orthologous genes. Lifestyles of the species used in this study are colour-coded for saprotrophic (blue) and opportunistic pathogenic (orange) species. The genome size for each genome assembly is indicated, with the respective proportions assigned as transposable elements (black), simple repetitive sequences (grey), and non-repetitive DNA (green). The genes predicted to be involved in carbohydrate transport and metabolism (red), and lipid transport and metabolism (yellow) and additional proteins (brown) are indicated for each strain. In the last column, the total number of detected tRNA genes (purple) is depicted.

opportunistic pathogens, these differences were not consistent among *Apiotrichum* and *Cutaneotrichosporon* species (Supplementary Fig. 4), suggesting different evolutionary histories among genera.

Given the importance of carbohydrate degradation for saprotrophic species and lipid metabolism in pathogenic species, we then assessed the gene composition of these pathways across saprotrophic and opportunistic pathogenic species, focusing on the common pathogenic *Cryptococcus*, *Cutaneotrichosporon*, and *Trichosporon* genera (Supplementary Data 3). However, functional annotation of orthologous genes exclusively detected in genomes of saprotrophic or opportunistic pathogens did not reveal discernible patterns associated with their respective lifestyles (Supplementary Data 4, 5). These findings suggest that other factors beyond gene content play a more critical role in determining fungal lifestyle within the *Trichosporonales* and *Tremellales* orders.

### tRNA composition and expansion are independent of lifestyle

Due to high tRNA variation among species (Fig. 1 and Supplementary Data 6), we explored the distribution and phylogenetic expansion of these genes (Supplementary Note 1). Our analyses revealed high variation in tRNA gene content (51 to 1455 genes) and an uneven distribution of tRNA genes across species (Supplementary Fig. 5). The anticodon types ranged from 39 to 46 per genome (Supplementary Fig. 6 and Supplementary Data 6), with most genomes containing tRNAs decoding all 20 universal amino acids, although some specific tRNAs were absent for some species, such as tRNAs decoding His or Asp (Supplementary Fig. 7 and Supplementary Data 7). The distribution of tRNA genes may be determined by the underlying phylogenetic relationships of species. In order to account for a putative phylogenetic signal in our analyses of tRNA gene composition, we estimated the phylogenetic signal by using Blomberg's K and associated

significance values (Supplementary Note 1). Despite the high phylogenetic signal of five genes, most of the tRNA gene expansion and composition (38 out of 45 genes) showed a reduced phylogenetic signal, potentially indicating multiple independent evolutionary events (Supplementary Fig. 8). In fact, we observe that the tRNA composition was similar among distantly related species (Supplementary Fig. 9), suggesting convergent evolution with respect to tRNA genes. Interestingly, we observed expansions of certain tRNA gene families across different species, yet the most frequently used synonymous codons did not consistently correspond with the most abundant tRNA gene families in the genomes, suggesting complex relationships between tRNA availability and codon usage patterns (Supplementary Data 8 and Supplementary Fig. 10).

To gain insights into the observed tRNA expansion (Fig. 1, Supplementary Data 6 and Supplementary Fig. 7) we further analysed the intragenomic sequence variation among multiple-copy tRNA genes (Supplementary Note 2, 3). Hereby, we found that the nucleotide variation within each gene family (i.e., genes with the same anticodon sequence) was inversely proportional to gene copy number (Supplementary Fig. 11A). High-copy tRNA genes displayed low sequence variation, while low-copy genes exhibited high divergence, suggesting recent expansion for the most abundant families (Supplementary Fig. 11B). Within each genome, when multiple tRNA genes decoded the same amino acid, at least one showed higher conservation (Supplementary Figs. 12, 13). We speculate that these patterns reflect different selective pressures among tRNA genes, both within and across amino acid groups, which also differed among isolates (Supplementary Fig. 14).

We next addressed whether tRNA composition and signatures of tRNA expansion could reflect functional adaptation to distinct lifestyles. To test this, we investigated if the content of tRNA genes differed among fungal species with different predicted fungal lifestyles.

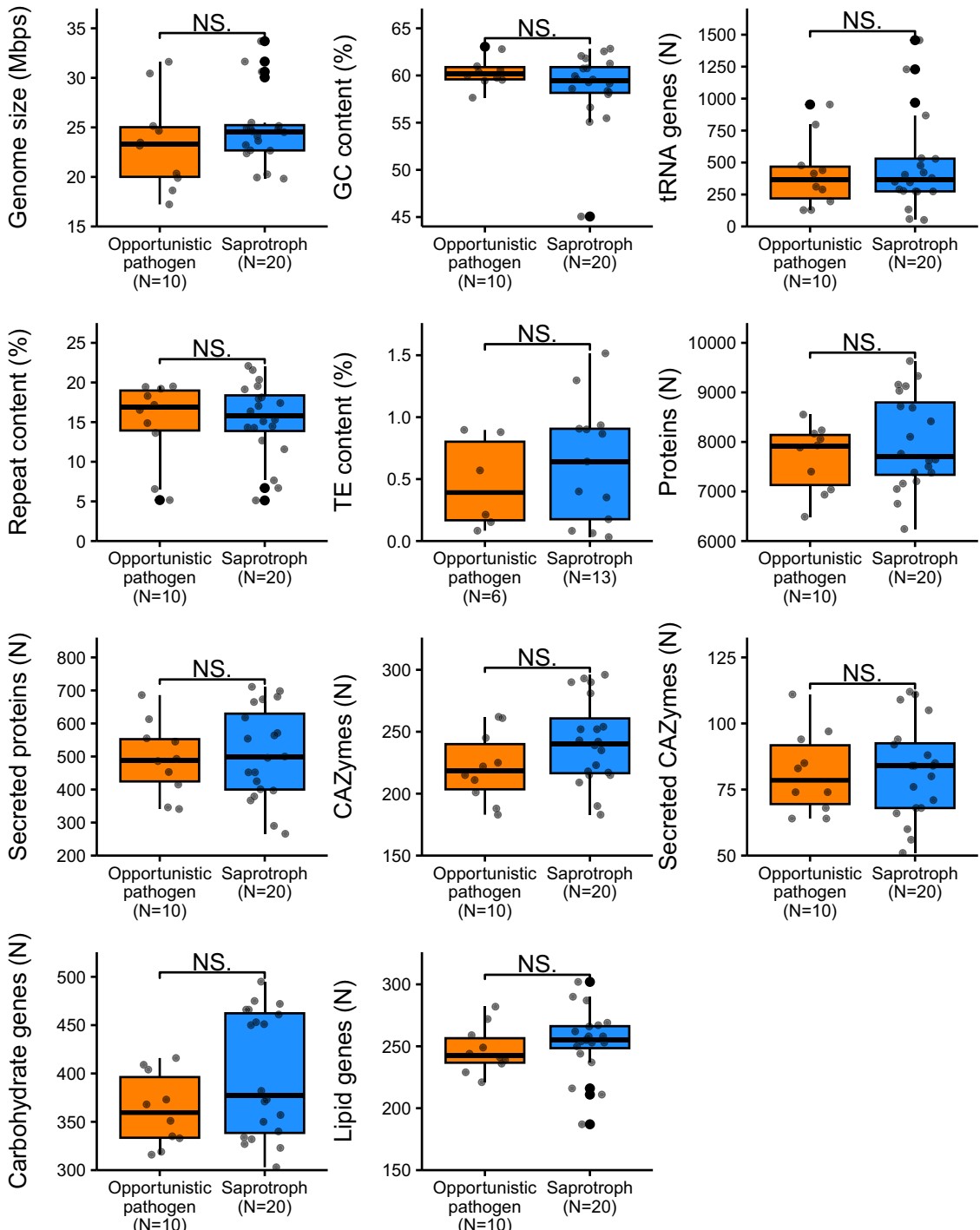

**Fig. 2 | Comparison of genomic features between opportunistic pathogenic and saprotrophic lifestyles among species in the *Trichosporonales* order.** No significant statistical differences ("NS." two-sided Wilcoxon signed-rank test *P*-value ≥ 0.05) were observed. To avoid overestimation, only 1 genome per species was considered, and the total number of genomes included per category is indicated. Secreted proteins and secreted CAZymes contained a signal peptide. Carbohydrate and lipid genes include genes involved in transport and metabolism. Each boxplot indicates the median value (centre line), the 25th and 75th percentiles (lower and upper hinges), and the most extreme data points (whiskers) within 1.5 × inter-quartile range. Individual points beyond the whiskers are outliers.

Our results indicated that tRNA content was indistinguishable between species with opportunistic pathogenic and saprotrophic lifestyles (Fig. 2 and Supplementary Data 2). However, tRNA content was strongly correlated with genome size (Supplementary Fig. 15A) among all *Trichosporonales* (Spearman's Rho = 0.75, *P* = 6.5e − 06) and among opportunistic human pathogenic (Spearman's Rho = 0.92, *P* = 0.001) and saprotrophic (Spearman's Rho = 0.58, *P* = 0.01) species, suggesting a strong link between tRNA gene content expansion and genome size.

Moreover, we observed a correlation between TE content and tRNA gene content (Spearman's Rho = 0.64, *P* = 6.7 × 10$^{-3}$), indicating that TEs also may play a role in the diversification of tRNAs (Supplementary Fig. 15B).

Overall, our results highlight a dynamic tRNA repertoire to some extent defined by the phylogenetic relationship of the species, but also by different selective forces, and recent expansion events independent of lifestyle.

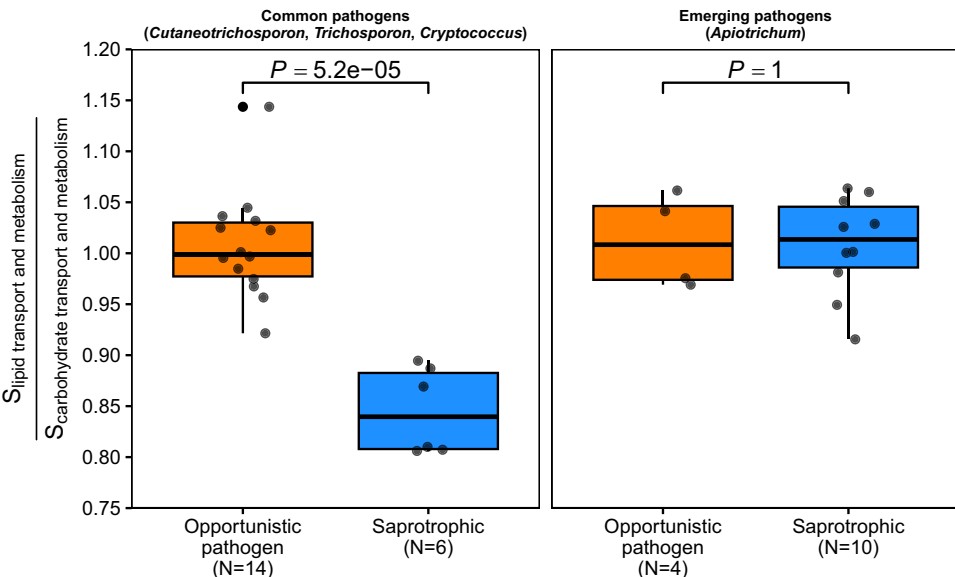

**Fig. 3 | Relative codon optimisation for genes involved in lipid or carbohydrate transport and metabolism.** The relative codon optimisation (S ratio) is compared between genera comprising both opportunistic pathogenic and saprotrophic species, except for *Trichosporon*. The opportunistic pathogens were grouped into common (*Cutaneotrichosporon*, *Trichosporon*, and *Cryptococcus*) or emerging (*Apiotrichum*) pathogens. Comparisons for each genus are provided in Supplementary Fig. 4. In order to consider intraspecific variation, the analysis considers several genomes for some species. The results considering 1 genome per species are consistent with the ones presented and are shown in Supplementary Fig. 18. Each boxplot indicates the median value (centre line), the 25th and 75th percentiles (lower and upper hinges), and the most extreme data points (whiskers) within 1.5 × inter-quartile range. Individual points beyond the whiskers are outliers.

## Relative codon optimisation of metabolic pathways predicts fungal lifestyle

We subsequently investigated whether tRNA composition influences the translation of genes associated with lifestyles. Based on the tRNA composition of each genome and codon usage in genes, we inferred the translation efficiency for genes associated with carbohydrate and lipid metabolic pathways, which are essential for growth in the environment as saprotrophs and in association with a human host as opportunistic pathogens, respectively. To infer the translational adaptation for these two lifestyles, we predicted the codon optimisation index (S) for lipid and carbohydrate transport and metabolism pathways[25]. The parameter S and respective statistical significance were estimated based on the tRNA gene pool, codon usage bias, and effective number of coding codons among all genes in the respective metabolic pathway. Therefore, S was used as a proxy for the measure of selection on the overall translational efficiency within these pathways. To account for inter-specific variation in codon usage bias, S values were normalised by comparing the mean S values across genes associated with lipid and carbohydrate metabolic pathways (S ratio) or by normalising pathway-specific S values to the genome-wide mean S (S normalised) (see Materials and Methods). A higher relative codon optimisation was interpreted as a proxy improved translation efficiency.

First, we normalised the codon optimisation (S) values against the genome-wide S for all genes and compared the normalised S across pathways involved in metabolism, cellular processes and signalling, and information storage and processing between saprotrophic and common opportunistic pathogenic species with clearly defined lifestyles (*Cryptococcus*, *Trichosporon*, *Cutaneotrichosporon*). Most pathways showed no significant differences (Wilcoxon test $P > 0.05$) in the normalised S values, except for the lipid transport and metabolism (Wilcoxon test $P = 0.0068$), cellular motility (Wilcoxon test $P = 0.0039$) and post–translational modification, protein turnover, and chaperones (Wilcoxon test $P = 0.046$) functional categories (Supplementary Fig. 16, Supplementary Data 2 and Supplementary Data 9). These results indicate significantly more optimisation in genes involved in lipid metabolism in opportunistic pathogens compared to saprotrophs and may reflect a key adaptation related to pathogen lifestyle.

In addition, we found that the absolute S and the normalised S values were similar for genes involved in both lipid and carbohydrate metabolic pathways (Wilcoxon test $P = 0.77$ and $P = 0.80$, respectively) in opportunistic pathogenic species (Supplementary Fig. 17). However, S was significantly higher for carbohydrate metabolism than for lipid metabolism (Wilcoxon test $P = 0.041$ and $P = 0.002$) in saprotrophic species. Moreover, the relative optimisation of the lipid in relation to carbohydrate metabolic pathways was significantly different (Wilcoxon test $P = 5.2 \times 10^{-5}$) between lifestyles (Figs. 3, 18 and Supplementary Data 2). The direct comparison of single-copy orthologous genes detected in all species further suggested higher normalised S values (Wilcoxon test $P = 0.046$) in genes related to the lipid metabolism in opportunistic pathogens compared to saprotrophic species (Supplementary Fig. 19 and Supplementary Data 10).

In order to test the predictive value of the S ratio (S lipid: S carbohydrates), we performed a permutation test and trained a decision tree classification model[34] using the *Cryptococcus*, *Trichosporon* and *Cutaneotrichosporon* species. Based on the permutation test, the S ratio was able to significantly explain lifestyle (empirical *P*-value = 0.008). In addition, the decision tree model trained on the full dataset classified the lifestyle accurately for all isolates (100% accuracy). To further test model robustness, we randomly sampled four saprotrophs and ten pathogens to train a model and used the remaining one saprotroph and three pathogens to test the model. In this cross-validation approach, all pathogenic species were correctly predicted (100% accuracy). For the saprotrophic species, the accuracy of the model was of 80%, due to one species (*Cryptococcus amylolentus*) being consistently misclassified due to the S ratio being near the classification threshold (Supplementary Data 11). The remaining saprotrophic species were correctly classified (100% accuracy). Using this decision tree approach, we conclude that the S ratio is a suitable predictor of fungal lifestyle in the studied species, particularly for distinguishing opportunistic pathogens from saprotrophic species.

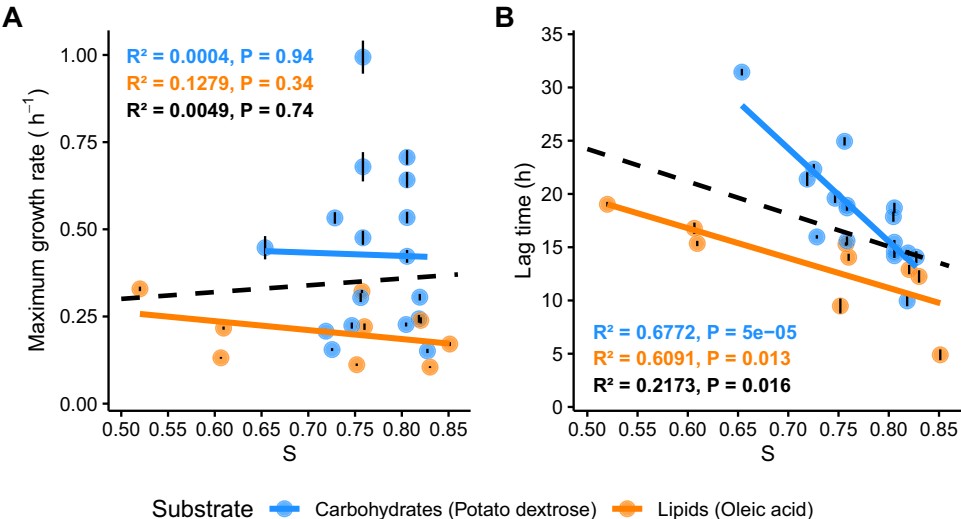

**Fig. 4 | Fitness of fungal isolates in carbohydrate-rich and lipid-rich media in relation to their codon optimisation (S).** The maximum growth rate (**A**) and the lag time (**B**) are plotted against the S of the corresponding metabolic pathway (carbohydrate or lipid metabolism). The correlation for each pathway is colour-coded (blue, carbohydrates; orange, lipids), while the overall correlation of both datasets is indicated by the black dashed line. All correlations are derived from a linear regression model. Data are presented for isolates with consistent growth patterns across four independent replicates. Error bars represent the uncertainty in the estimated growth rate and lag time from the model fit for each replicate, as determined by *omniplate*. Isolates exhibiting abnormal growth or inconsistent replicates were excluded from the analysis.

Altogether, these results indicate that saprotrophic species with a pathogenic lifestyle (i.e., opportunistic pathogens) have similar codon optimisation for both carbohydrates and lipid metabolic pathways, while exclusively saprotrophic species have higher codon optimisation for carbohydrate metabolism (Supplementary Fig. 17). Lifestyles were distinguishable based on the relative codon optimisation (S ratios) of metabolic pathways, which may be relevant to survival in the respective habitats (Fig. 3 and Supplementary Data 2).

These findings indicate selection at the translational level reflected by a significantly higher translational efficiency for carbohydrate-related metabolic genes in saprotrophic species and support the hypothesis that lifestyle adaptations are imprinted in the codon optimisation of lifestyle-relevant pathways.

The emerging opportunistic pathogenic *Apiotrichum* species (*A. mycotoxinovorans* and *A. veenhuisii*) showed a similar codon optimisation between lipid and carbohydrate pathways and were indistinguishable from other well-known opportunistic pathogens (*Cryptococcus* spp., *Cutaneotrichosporon* spp., and *Trichosporon* spp.) (Supplementary Figs. 17, 18, 20). In addition, *Apiotrichum* species classified as saprotrophic (e.g., *A. gracile* or *A. porosum*) showed a similar adaptation for both pathways and were also indistinguishable from other pathogens (Wilcoxon test *P* = 1, Fig. 3 and Supplementary Fig. 18). This is in contrast to the patterns observed for other genera, where common opportunistic pathogens and their saprotrophic closely related species exhibited clear differences in translational adaptation. The similarity between saprotrophic and pathogenic *Apiotrichum* species suggests that some species currently considered non-pathogenic may represent potential human pathogens that have remained undetected or misidentified in clinical settings.

### Codon optimisation of metabolic pathways predicts fitness
Subsequently, we investigated whether predicted translational adaptation impacted the fitness of the species. We performed growth assays in carbon-rich (potato dextrose) and lipid-rich (oleic acid-based) media. To estimate the fitness effect, we correlated growth parameters with the codon optimisation (S) of the carbohydrate and lipid metabolic pathways, respectively. The maximum growth rate and lag time were used as estimators of fitness for 21 fungal isolates representing 16 species (Supplementary Data 12). The maximum growth rate corresponds to the highest speed at which the cell density increases, while the lag time refers to the time required for the fungal cells to adjust to the surrounding environment, often associated with changes in transcriptome and proteome, before starting exponential growth[35].

In carbohydrate-rich media, we observed the highest maximum growth rate for *Cutaneotrichosporon dermatis* TS-012 (0.99 h$^{-1}$, standard error 0.05 h$^{-1}$) and the lowest for *Apiotrichum laibachii* TS-076 (0.15 h$^{-1}$, SE 0.006 h$^{-1}$). In lipid-rich media, *Cutaneotrichosporon cyanovorans* TS-095 and *Trichosporon asahii* TS-108 displayed the highest (0.33 h$^{-1}$, SE 0.006 h$^{-1}$) and the lowest (0.10 h$^{-1}$, SE 0.003 h$^{-1}$) maximum growth rates, respectively. *Apiotrichum akiyoshidainum* TS-001 had the shortest lag phase (9.96 h, SE 0.50 h) in carbohydrate-rich media, while *Cutaneotrichosporon curvatus* TS-039 had the longest (31.43 h, SE 0.31 h). In lipid-rich media, *Apiotrichum laibachii* TS-076 had the shortest lag phase (4.90 h, SE 0.51 h), while *Cutaneotrichosporon cyanovorans* TS-095 had the longest (19.03 h, SE 0.23 h).

Our analysis revealed no significant correlation (*P*-value > 0.34) between codon optimisation (S) and maximum growth rate in either substrate (Fig. 4A). However, we observed a significant correlation (R$^2$ > 0.6, *P*-value < 0.013) between lag time and *S*-values (Fig. 4B), indicating that isolates with higher S of the corresponding metabolic pathway displayed shorter lag phases. These results suggest that a higher codon optimisation of metabolic pathways may provide a fitness advantage by allowing faster adaptation to the surrounding environment and quicker onset of growth.

### Saprotrophic species grow at mammal body temperatures
The ability to grow at high temperatures is a critical virulence factor for opportunistic pathogens of humans and other mammal species. Therefore, we further assessed the ability of isolates to grow at 33 °C and 37 °C. The isolates represented opportunistic pathogenic and saprotrophic species. Most opportunistic pathogenic species (6 isolates) exhibited growth at 33 °C and 37 °C (Fig. 5) except for *Cutaneotrichosporon curvatus* TS-039, which was only able to grow up to 28 °C (Supplementary Figs. 21, 22). Most saprotrophic species were unable to grow above 28 °C (Supplementary Fig. 21); however, we

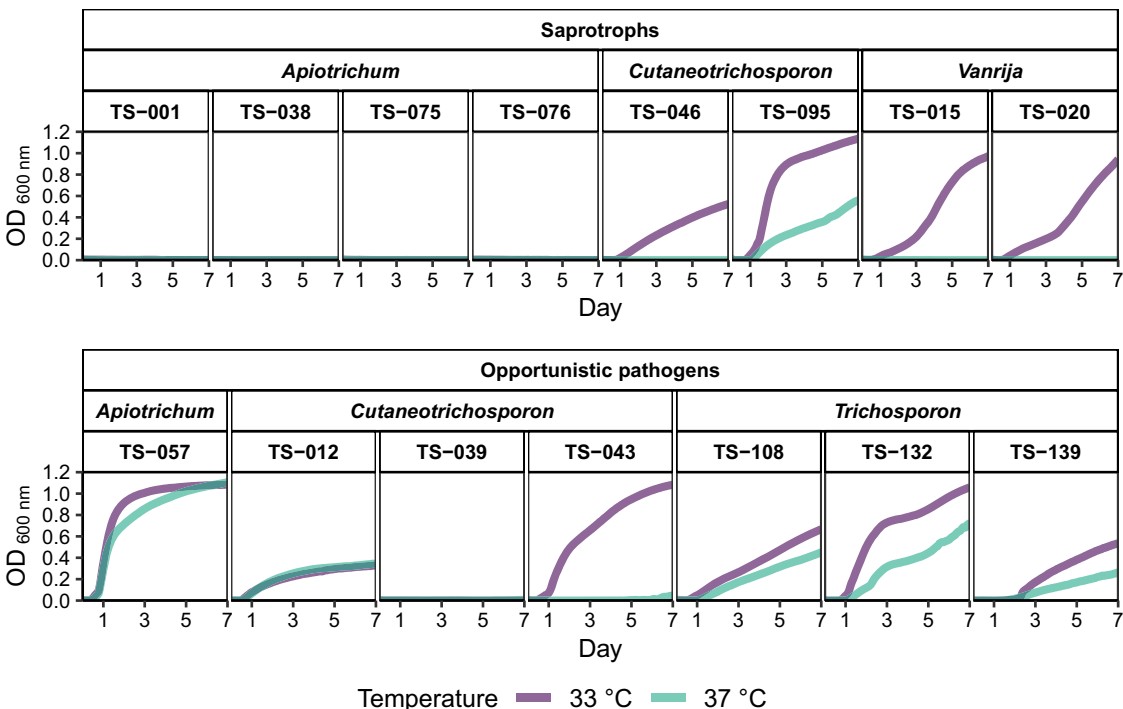

**Fig. 5 | Growth curves of isolates cultivated at 33 °C and 37 °C for 7 days.** The optical density (OD) values at 600 nm are presented, excluding the baseline OD of the growth medium. The isolates are separated according to lifestyle and genus. Temperatures are colour-coded (33 °C, purple; 37 °C, cyan). Isolates that show no growth only grow at temperatures up to 23 °C or 28 °C (Supplementary Figs. 21, 22).

observed growth of 4 isolates at 33 °C (*Vanrija humicola* TS-015 and TS-020, *Cutaneotrichosporon oleaginosus* TS-046, and *Cutaneotrichosporon cyanovorans* TS-095) and 1 isolate at 37 °C (*Cutaneotrichosporon cyanovorans* TS-095) (Fig. 5).

These results suggest that growth at high temperatures is not a sole determinant of pathogenicity, since some pathogenic species contain strains not growing at high temperatures, like *Cutaneotrichosporon curvatus* TS-039 and *Cutaneotrichosporon dermatis* TS-043 growing at 28 °C and 33 °C, respectively (Supplementary Fig. 22). At the same time, some saprotrophic species were able to grow above 33 °C (e.g., *Cutaneotrichosporon oleaginosus* TS-046). We speculate that adaptation to higher temperatures may play a role in ecological niche expansion and ability to infect human hosts, however, it is likely one of multiple factors involved in pathogenicity.

Some saprotrophic species (e.g., *Vanrija* spp.) showed similar relative codon optimisation patterns to those of opportunistic human pathogens (Fig. 3 and Supplementary Fig. 20). The growth temperature experiments (Fig. 5, Supplementary Figs. 21, 22) complement these findings by showing that some saprotrophic species are able to grow at mammal body temperature (e.g., *Vanrija* spp.), similar to opportunistic human pathogens. Together, these results suggest that some saprotrophic species may have the thermo- and translational adaptation associated with pathogenicity, further suggesting that these may emerge as potential human opportunistic pathogens.

## Discussion

In this study, we provide a comprehensive overview in *Tremellomycetes*, including several emerging pathogens, of the evolution of genome composition and tRNA diversity and the resulting implications on the emergence of pathogenicity. Our results suggest that the emergence of pathogenicity may be associated with codon optimisation for lipid metabolism. We further show that some species with an optimised codon composition for lipid metabolism can grow at higher temperatures. Taken together, our findings reflect adaptations relevant for infection of mammalian hosts.

In this study, we used different representations of codon optimisation (S) to capture complementary aspects of translational adaptation. Absolute *S*-values directly reflect the co-adaptation between codon usage bias and the genomic tRNA pool within each genome, while also integrating the effective number of codons and the GC content at the third position. Since individuals and species often show variations in the tRNA pool and codon usage biases, absolute *S*-values may not be directly comparable across genomes. To account for different genomic backgrounds, we calculated normalised *S*-values, which express the codon optimisation of a given pathway relative to the genome-wide *S*-value (S absolute pathway 1 : S absolute genome). This metric indicates whether a pathway is more or less optimised than expected within the same genome. Finally, S ratios provide a direct comparison between two functional categories within a genome (S absolute pathway 1 : S absolute pathway 2), which reflects whether a pathway is more strongly optimised relative to another. Altogether, we consider the three metrics to provide complementary insights: absolute *S*-values assess the global level of translational adaptation in a genome, normalised *S*-values emphasise deviations from the genomic baseline, and S ratios determine relative optimisation between pathways. By combining these metrics, we assess the contribution of codon optimisation to fungal lifestyle adaptation.

The distribution of opportunistic pathogenic species across the phylogeny (Fig. 1) suggests that pathogenicity, or the ability to colonise human and animal hosts, has emerged multiple times independently in the *Tremellomycetes*, during the evolution and diversification of species in the two major orders *Trichosporonales* and *Tremellales*. Genome duplication and hybridisation are major evolutionary events that are often connected with pathogenicity[36]. In the *Trichosporonales*, 3 interspecies hybrid genomes have been reported, all of which belong to known pathogenic species (*Trichosporon coremiiforme*, *Trichosporon ovoides*, and *Cutaneotrichosporon mucoides*)[31,32]. The detection of hybrid genomes belonging to pathogenic species supports the hypothesis that the emergence and speciation of fungal pathogens is often associated with hybridisation[37]. In our study, we detected 1

additional genome containing a high number of duplicated genes, indicating a possible genome duplication or hybridisation event, belonging to a strain of the pathogen *Cutaneotrichosporon dermatis* (Supplementary Fig. 1). The genomic variation, as well as highly diverse ecology, suggest that this order might possess a previously unreported high species diversity. Comprehensive biodiversity assessments and taxonomic studies are needed to clarify species diversity in this group. Such studies are crucial for epidemiological studies in clinical settings, since closely related species commonly have different virulence and susceptibility to antifungal drugs[38].

Genome evolution and translational regulation can be linked to pathogenicity and the ability of pathogenic species to colonise multiple hosts[19]. Our comparative genomic approaches suggest that several saprotrophic species share features with opportunistic pathogens (Fig. 2). Several saprotrophic species (*Apiotrichum*, *Haglerozyma*, *Pascua*, *Prillingera*, and *Vanrija*) show similar adaptation to lipids as some species reported to be dermatophytes and opportunistic pathogens (*Apiotrichum*, *Cutaneotrichosporon*, *Trichosporon*, and *Cryptococcus*) (Fig. 3, Supplementary Figs. 4, 20). In addition, some species display the ability to grow at human body temperatures (Fig. 5 and Supplementary Fig. 21). This raises the concern that some species currently classified as strictly saprotrophic may evolve to become opportunistic pathogens.

Substrate specialisation is ecologically relevant in the context of the transition from saprotrophic to pathogenic lifestyles. The opportunistic pathogen *Cryptococcus deneoformans* shows an increased expression of lipid-related genes during infection[39,40]. This further suggests that in mammal hosts, lipid-rich environments may select for opportunistic pathogens with optimised lipid metabolism. Our experimental data link codon optimisation in genes related to certain metabolic pathways to increased fitness in the respective substrates. We propose that codon optimisation and improved translation efficiency represent a mechanism of faster adaptation (Fig. 4). Physiological adjustment time (here determined as the lag-time) was more strongly reduced in carbohydrate-rich conditions than in lipid-rich conditions, which may be influenced by the codon optimisation in the corresponding metabolic genes. These results suggest that codon optimisation patterns could serve as potential predictive genomic markers for fungal lifestyles. In fungi, the carbohydrate metabolism often involves redundant and multifunctional proteins, while lipid metabolism relies mainly on functionally specific proteins[41,42]. We speculate that the codon optimisation in genes involved in carbohydrate metabolism may involve a multitude of subprocesses. In addition, the energy released during carbohydrate metabolism is more immediately available than for lipid metabolism, which requires additional processing. Hence, we propose that the difference in translation efficiency for carbohydrate and lipid metabolism may reflect a difference in the complexity of the underlying cellular processes.

Based on our comparative genome analysis and our in-vitro fitness assays (Fig. 4), our results indicate that adaptive evolution may shape the composition of tRNA genes and codons to tune the expression of genes associated with distinct lifestyles[24]. Although we did not analyse gene expression in our study, previous studies on *Cryptococcus deneoformans* show that genes involved in carbohydrate and lipid metabolism are highly expressed during infection and enriched in human cerebrospinal fluids[43], supporting our hypothesis that these gene sets are important for pathogenesis. Optimisation of protein translation may greatly impact the capacity of an organism to cope with certain environmental conditions. For example, in human cells, the translation rate of genes with optimal codon usage is 58% faster (4.9 codons/s) than non-optimised genes (3.1 codons/s)[44], emphasising the role of codon optimisation in translational speed.

Gene expression and regulation are directly affected by several molecular mechanisms, including chromatin remodelling, epigenetic modifications, transcription factor binding, and the use of alternative promoters[45,46]. Transcriptional reprogramming can be linked to shifts between saprotrophic and host-associated lifestyles, for example, via the upregulation of stress-response pathways, secreted enzymes, or virulence factors[47,48]. Hence, transcription regulation defines the repertoire of expressed genes, while translational adaptation regulates the protein synthesis efficiency to meet specific requirements. These mechanisms suggest that ecological transitions are enabled by both transcriptional regulation and translational adaptation. While we show that translational adaptation provides an advantage in niche adaptation (Fig. 4), future studies addressing the relevance of transcriptional regulation and how this interacts with translation efficiency will be crucial to fully understand the molecular mechanisms involved in lifestyle transitions.

Several *Trichosporonales* species have been previously isolated as commensals from the healthy skin of mammals[49]. In pathogenicity tests, these isolates caused successful skin infections and invasive infections in immunosuppressed mice, and most species caused significant damage to the liver and skin of healthy immunocompetent mice[49]. Another study showed that the human opportunistic pathogen *Tr. asahii* is capable of infection of immunocompetent mice and dissemination into their kidneys, liver, spleen, and brain[50]. Infection mechanisms appear to be shared among different genera, as pathological characteristics are similar when mice are infected by different species and genera[49]. This further supports our hypotheses that saprotrophic *Apiotrichum*, *Pascua*, and *Vanrija* species could represent potential human opportunistic pathogens, as suggested by their adapted translation to lipid metabolism (Fig. 3 and Supplementary Fig. 20) and mammalian body temperature, which is comparable to that of pathogens (Fig. 5 and Supplementary Fig. 21). Pathogenicity tests and clinical reports also indicate that this order has a broad range of hosts, including insects and mammals such as humans, mice, dogs, and cats[51-53].

Our analyses link genomic information with fungal lifestyle across the *Tremellomycetes*. We detected expansion of the translation machinery (i.e., tRNAs) independently of lifestyle (Supplementary Note 1–3 and Supplementary Discussion), while the translation efficiency of metabolic genes showed signatures of adaptation (i.e., codon optimisation) linked to lifestyle. We found evidence for an evolutionary scenario in which distinct habitats select for the optimal translation of genes that are involved in the successful colonisation of the respective habitat. Our results also suggest that because saprotrophic species are indistinguishable from opportunistic pathogens based on many genomic features, these fungi might have the potential to evolve towards pathogenic lifestyles. We predict that lifestyles are not only strictly defined by gene repertoires and expression profiles, but also by translational adaptation of relevant pathways reflected by the translation efficiency of these gene sets. Our study provides the basis for future research addressing evolutionary gene regulation, translational adaptation, and dynamic lifestyles in fungi.

The similarities observed in many genomic features between saprotrophic species and opportunistic pathogens raise concerns about the potential for some saprotrophic species to become opportunistic human pathogens. This emphasises the importance of developing tools to accurately classify species based on their ecological strategies using genomic and physiological data, because using only species occurrence data to classify lifestyles may not be sufficient. Comparative genomic studies can help to predict the risk of future fungal infection outbreaks by identifying species with genomic signatures similar to already known pathogens. Furthermore, the coevolution of tRNA genes and codon composition can be linked to the translational regulation of genes under different environmental conditions[21], which may have implications for the ability of pathogenic species to colonise different hosts[19]. A deeper understanding of evolutionary processes associated with lifestyle adaptation is necessary for developing effective strategies to control fungal infections.

## Methods

### Genomic data acquisition

Assembled genomic data of 50 fungi were downloaded from the NCBI database, the US DOE Joint Genome Institute's MycoCosm database (https://mycocosm.jgi.doe.gov), or ATCC Genome Portal (https://genomes.atcc.org/genomes). A total of 5 hybrid or diploid genomes were excluded from further analyses based on literature[31] and duplicated BUSCO content (Supplementary Fig. 1). Strains with a predicted proteome completeness > 85% and duplicated BUSCOs < 5% were selected for further analyses. The final dataset comprised 45 haploid genomes, representing 30 species from the *Trichosporonales* order and 4 *Cryptococcus* species from the sister-order *Tremellales* as outgroups.

Species were considered to be opportunistic human pathogens if 1) they were previously reported in human clinical settings and 2) the identity was confirmed with molecular markers (Supplementary Data 1). Species without clinical reports were considered to be saprotrophic.

### Coding gene prediction

To avoid gene prediction biases, all assembled genomes were processed using the funannotate pipeline v1.8.10 with the same settings[54]. The pipeline included size filtering, cleaning, sorting, and softmasking steps with default parameters, followed by gene prediction using "cryptococcus" as *augustus_species* and *busco_seed_species* and "basidiomycota" as *busco_db*. Genome completeness was predicted using BUSCO v5.3.2 in comparison to the Basidiomycota lineage database (basidiomycota_odb10)[55]. For gene comparisons, only coding sequences (CDS) without a predicted stop codon in the inner sequence were considered for further analyses.

### Phylogenetic reconstruction

Orthologous protein sequences among the studied *Trichosporonales* were predicted by using OrthoFinder v2.5.4[56]. A set of 1438 single-copy orthologous genes present in all species were obtained and subsequently individually aligned with MAFFT v7.505[57]. The individual gene alignments were concatenated and used to calculate a maximum likelihood phylogenetic tree with RAxML v.8.2.12 using 500 bootstrap replicates, the PROTGAMMAWAG model, 123 as seed number for the parsimony inferences, and a random seed of 321[58].

### tRNA gene prediction and analyses of sequence variation

tRNA genes were predicted with tRNAscan-SE 2.0.9 with default parameters for eukaryotic organisms[59]. To ensure the high quality of gene predictions, all possible pseudogenes and tRNA genes with a covariance score < 30 were excluded from further analyses[59]. In addition, a subset of tRNA genes were manually curated based on their predicted structure, which validated the quality-filtering criteria, and all curated and quality-filtered genes were subsequently retained. Studies have empirically demonstrated that in fungi and bacteria, the tRNA copy number in the genome correlates with the abundance of each tRNA family in the cell[60–63]. Therefore, gene copy number in the genome was used as a proxy for the available tRNA pool in the cell.

To explore the overall evolutionary dynamics of tRNA genes, including gene duplications and expansion events, we analysed the full-length tRNA gene sequences, including introns. To estimate sequence variation, the complete tRNA gene sequences were obtained for each genome and aligned for each anticodon type using MAFFT v6.864b[57]. Only genes with ≥ 2 copies were considered for intragenomic sequence variation analyses. To estimate intragenomic sequence variation for each tRNA gene family (i.e., anticodon type), the genetic distance was predicted based on sequence alignments using the Kimura-2-parameter distance model[64] in the *ape* R package v5.6-2[65]. Outliers with a distance > 10 were excluded, which likely resulted from poor alignments.

### Prediction of additional genomic features

Gene functions and pathways were predicted by eggNOG v2.0.8[66] (Supplementary Data 14). To predict CAZymes, proteomes were scanned against the dbCAN2 v10.0 database[67], using HMMER v3.3[68]. Matches with an e-value < 1e-15 and a coverage > 0.35 were used for further analyses. Secreted proteins were predicted from each proteome with SignalP 6.0 g[69]. Transposable elements were predicted by analysing each genome assembly with REPET3 pipeline[70,71], and according to previous studies[72]. Possibly due to sequencing technologies or assembly methods, the REPET3 approach was unable to predict TEs for some genomes. Therefore, the analyses regarding TEs consider a lower number of species. Repetitive elements were determined by funannotate during softmasking with tantan[73].

### Codon optimisation and tRNA adaptation index estimation

To assess potential translational selection, the tRNA adaptation index (tAI) and codon optimisation index (S) were calculated with the *tAI* v0.2.1 R package[25]. Here, we consider tAI as a proxy for translation efficiency and S as an estimate of the strength of translational selection, respectively. The tAI measures the co-evolution between a given CDS and the respective genomic tRNA pool[25,74]. The tAI is calculated based on the tRNA gene family count, and the codon count and the distribution per gene in the dataset. The S index estimates the intensity of translational selection based on the correlation between the tAI and the effective number of codons (corrected for GC content at third codon positions)[25,74]. The statistical significance of the S parameter (the correlation between the tAI and the adjusted codon count) was tested by a Monte Carlo test from the same R package[25]. The S index, therefore, reflects the intensity of translational selection on gene sets and respective translation efficiency. The synonymous relative codon usage was calculated for the CDS dataset with BioKIT v0.1.1[75].

The lipid and carbohydrate transport and metabolism-related genes were defined according to the narrowest orthologous groups' category. The tAI was calculated for each individual gene, while the S was calculated for sets of genes. The normalised translation efficiency related to lipids and carbohydrates was defined as the ratio between the S of the gene sets predicted to be involved in the transport and metabolism in these two metabolic pathways ($S_{lipid\ transport\ and\ metabolism}$ : $S_{carbohydrate\ transport\ and\ metabolism}$). This ratio was used to address different levels of optimisation between genomes and compare the relative adaptive translation and efficiency of these metabolic pathways within each genome. This ratio indicates a similar level of adaptation to both lipid and carbohydrates (ratio ≈ 1) or a selective translational optimisation towards genes involved in carbohydrate (ratio < 1) or lipid (ratio > 1) transport and metabolism. In addition, the *S*-value was normalised (S norm) for each pathway in relation to the genomic *S*-value, to allow direct comparison between pathways among species. A higher codon optimisation in a given pathway, relative to another pathway or to the genomic background, may reflect functional prioritisation during translation.

To test the predictive value of the S ratio (S lipid : S carbohydrates) for fungal lifestyles, species from the genera *Cryptococcus*, *Trichosporon*, and *Cutaneotrichosporon* were used in a permutation test and for training a decision tree classification model[34]. These tests were performed with the *C5.0* v0.2.0[76] R package. A permutation test (*n* = 9999) was performed by randomly shuffling lifestyle labels to evaluate the statistical significance of the observed classification accuracy. Furthermore, a decision tree classifier (C5.0) was trained using the S ratio as the only predictor. To assess model robustness, 75% of the data were randomly selected by lifestyle for training (5 saprotrophs and 11 pathogens), and the remaining 25% (1 saprotroph and 3 pathogens) were used for testing. This procedure was repeated over 9999 iterations, and the classification probabilities and accuracies were recorded for each species.

## Phylogenetic signal estimation

The strength and direction of trait evolution was estimated based on Blomberg's K using phylogenetic distance as the only predictor of trait similarity among species[77]. Blomberg's K infers stabilising selection ($K > 1$) and neutral ($K = 1$) or convergent ($K < 1$) evolution of a trait relative to the expected variation under the Brownian Motion model of evolution[77]. This parameter was calculated using the *phytools* v1.0-3 R package using 100 permutations[78]. Only K values with $P < 0.05$ were considered significant. A previously calculated maximum likelihood phylogeny with branch length was used for the calculation (see above).

The influence of both phylogeny and lifestyle on the variance observed across different genomic datasets was tested by decomposing the phylogenetic structure into principal components and analysing them alongside lifestyle using a PERMANOVA approach[79]. To test for the correlation between tRNA gene composition and phylogenetic distance, the pairwise Bray-Curtis dissimilarity among all genomes was calculated using the *vegan* v2.6-4 R package, considering the copy number of each tRNA gene family gene[80]. The pairwise branch length among all genomes was calculated using the *phytools* v1.0-3 R package and was used as a measure of phylogenetic distance[78].

## Statistical analyses

Statistical analyses were performed in R v4.2.0[81] and RStudio 2022.02.2[82]. Data were visualised with *ggplot2*[83]. For correlative analyses, only 1 strain of each species was used (Supplementary Data 1). Wilcoxon tests were performed with the *ggsignif*[84] and *broom*[85] R packages. Linear regression models, *R*-values, and *P*-values were calculated with the *ggpmisc* R package[86]. Phylogenetic independent contrasts (PICs) were performed with the *ape*[65] R package.

## Yeast strains and media

The yeast cultures used in this study (Supplementary Data 12) were obtained from the Infrastructural Mycosmo Centre and Microbial Culture Collection Ex (EXF), German Collection of Microorganisms and Cell Cultures (DSMZ), Agricultural Research Service Culture Collection, USA (NRRL) and Westerdijk Fungal Biodiversity Institute (CBS). The isolates *Apiotrichum porosum* TS-027 and *Apiotrichum mycotoxinovorans* TS-057 were kindly provided by Víctor M. Flores-Núñez (University of Kiel, Germany) and Till Friedrich Schäberle (Fraunhofer IME, Germany), respectively. All isolates were grown on potato dextrose (PD) medium at 23 °C and stored at −80 °C. PD broth medium contained 24 g/L potato dextrose (Sigma-Aldrich). PD agar medium contained 39 g/L potato dextrose agar (Difco) and 10 mL/L 1 M TrisHCl (pH 8.0).

## Growth assays

All growth assays were performed on sterile 96-well microplates in a total volume of 200 μL. Growth was measured with an Epoch 2 plate reader using 600 nm absorbance. Measurements were performed every 30 mins until the stationary phase was reached (approximately 138–168 h). All media were inoculated with an initial optical density (OD$_{600 nm}$) of $2.5 \times 10^{-4}$ (approximately 100 – 1500 CFUs, Supplementary Data 13). Isolates were grown in quadruplicates. To determine maximum growth temperature, each isolate was pre-grown in PD broth medium for 3 days at 23 °C and then incubated at 33 °C and 37 °C in PD broth medium as detailed above. In addition, all isolates were grown at 18 °C, 23 °C, 28 °C, 33 °C and 37 °C in 200 μL as detailed above and in 5 mL in sterile reaction tubes.

To determine the fitness in carbohydrate- or lipid-rich media, isolates were pre-grown in PD broth or lipid-based media and then inoculated in the respective medium at 23 °C as detailed above. The lipid-based medium contained 2 mL/L oleic acid, 20 g/L ox bile, 10 mL/L Tween 40, and 6.7 g/L yeast nitrogen base without amino acids.

Maximum growth rate, lag time and respective estimated errors were determined from quadruplicates with omniplate v1.0[87].

## Reporting summary

Further information on research design is available in the Nature Portfolio Reporting Summary linked to this article.

## Data availability

Genome assemblies used in the manuscript are available under the following links: Apiotrichum akiyoshidainum HP2023 [https://www.ncbi.nlm.nih.gov/datasets/genome/GCA_002973495.1], Apiotrichum brassicae JCM 1599 [https://www.ncbi.nlm.nih.gov/datasets/genome/GCA_001600295.1], Apiotrichum_domesticum JCM 9580 [https://www.ncbi.nlm.nih.gov/datasets/genome/GCA_001599015.1], Apiotrichum gamsii JCM 9941 [https://www.ncbi.nlm.nih.gov/datasets/genome/GCA_001600315.1], Apiotrichum gracile JCM 10018 [https://www.ncbi.nlm.nih.gov/datasets/genome/GCA_001600335.1], Apiotrichum laibachii JCM 2947 [https://www.ncbi.nlm.nih.gov/datasets/genome/GCA_001600735.1], Apiotrichum montevideense JCM 9937 [https://www.ncbi.nlm.nih.gov/datasets/genome/GCA_001598995.1], Apiotrichum mycotoxinovorans CICC 1454 [https://www.ncbi.nlm.nih.gov/datasets/genome/GCA_013177335.1], Apiotrichum mycotoxinovorans GMU1709 [https://www.ncbi.nlm.nih.gov/datasets/genome/GCA_011290525.1], Apiotrichum mycotoxinovorans ACCC 20271 [https://www.ncbi.nlm.nih.gov/datasets/genome/GCA_001613755.1], Apiotrichum porosum JCM 1458 [https://www.ncbi.nlm.nih.gov/datasets/genome/GCA_001600255.1], Apiotrichum porosum DSM 27194 [https://www.ncbi.nlm.nih.gov/datasets/genome/GCF_003942205.1], Apiotrichum siamense L8in5 [https://www.ncbi.nlm.nih.gov/datasets/genome/GCA_023653615.1], Apiotrichum veenhuisii JCM 10691 [https://www.ncbi.nlm.nih.gov/datasets/genome/GCA_001600595.1], Cutaneotrichosporon arboriformis JCM 14201 [https://www.ncbi.nlm.nih.gov/datasets/genome/GCA_002335565.1], Cutaneotrichosporon curvatus JCM 1532 [https://www.ncbi.nlm.nih.gov/datasets/genome/GCA_001600275.1], Cutaneotrichosporon cutaneum JCM 1462 [https://www.ncbi.nlm.nih.gov/datasets/genome/GCA_001600715.1], Cutaneotrichosporon cyanovorans JCM 31833 [https://www.ncbi.nlm.nih.gov/datasets/genome/GCA_002335625.1], Cutaneotrichosporon daszewskae JCM 11166 [https://www.ncbi.nlm.nih.gov/datasets/genome/GCA_002335585.1], Cutaneotrichosporon dermatis JCM 11170 [https://www.ncbi.nlm.nih.gov/datasets/genome/GCA_003116895.1], Cutaneotrichosporon dermatis ATCC 204094 [https://genomes.atcc.org/genomes/e80264e2adb34f72], Cutaneotrichosporon oleaginosus ATCC 20509 [https://genomes.atcc.org/genomes/1bd8cbf8d02b479c], Cutaneotrichosporon oleaginosus ATCC20508 [https://www.ncbi.nlm.nih.gov/datasets/genome/GCA_008065305.1], Cutaneotrichosporon oleaginosus IBC0246 [https://www.ncbi.nlm.nih.gov/datasets/genome/GCF_001027345.1], Haglerozyma_chiarellii_ATCC_MYA-4694 [https://mycocosm.jgi.doe.gov/Trich1], Pascua_guehoae_JCM_10690 [https://www.ncbi.nlm.nih.gov/datasets/genome/GCA_001600415.1], Pascua guehoae Phaff 60-59 [https://mycocosm.jgi.doe.gov/Trigue1], Prillingera fragicola JCM 1530 [https://www.ncbi.nlm.nih.gov/datasets/genome/GCA_002335605.1], Trichosporon asahii ATCC 201110 [https://genomes.atcc.org/genomes/b7621150fd7849ee], Trichosporon asahii CBS 8904 [https://www.ncbi.nlm.nih.gov/datasets/genome/GCA_000299215.2, Trichosporon asahii JCM 2466 [https://www.ncbi.nlm.nih.gov/datasets/genome/GCA_001972365.1], Trichosporon asahii N5_275_008G1 [https://www.ncbi.nlm.nih.gov/datasets/genome/GCA_004026345.1], Trichosporon faecale JCM 2941 [https://www.ncbi.nlm.nih.gov/datasets/genome/GCA_001752585.1], Trichosporon inkin JCM 9195 [https://www.ncbi.nlm.nih.gov/datasets/genome/GCA_001752625.1], Vanrija humicola ATCC 9949 [https://genomes.atcc.org/genomes/781d93df71954299], Vanrija humicola CBS 4282 [https://www.ncbi.nlm.nih.gov/datasets/genome/GCA_008065275.1], Vanrija humicola JCM

1457 [https://www.ncbi.nlm.nih.gov/datasets/genome/GCA_001600 235.1], Vanrija humicola UJ1 [https://www.ncbi.nlm.nih.gov/datasets/genome/GCA_002897395.1], Vanrija pseudolonga DUCC4014 [https://www.ncbi.nlm.nih.gov/datasets/genome/GCA_020906515.1], Takashimella koratensis JCM 12878 [https://www.ncbi.nlm.nih.gov/datasets/genome/GCA_003116875.1], Takashimella tepidaria JCM_11965 [https://www.ncbi.nlm.nih.gov/datasets/genome/GCA_003116915.1], Cryptococcus amylolentus CBS 6039 [https://www.ncbi.nlm.nih.gov/datasets/genome/GCF_001720205.1], Cryptococcus deneoformans JEC21 [https://www.ncbi.nlm.nih.gov/datasets/genome/GCF_000091045.1], Cryptococcus floricola DSM 27421 [https://www.ncbi.nlm.nih.gov/datasets/genome/GCA_006352305.1], Cryptococcus gattii WM276 [https://www.ncbi.nlm.nih.gov/datasets/genome/GCA_000185945.1].

Strains with the code "DSMZ" are available at the German Collection of Microorganisms and Cell Cultures GmbH (DSMZ). Strains with code "EXF" are available at the Infrastructural Mycosmo Centre and Microbial Culture Collection Ex (EXF). Strains with code "NRRL" are available at the Agricultural Research Service Culture Collection (ARS-NRRL). Strains with code "CBS" are available at the Westerdijk Fungal Biodiversity Institute (CBS). The strain 'ac123' (TS-027) is physically available and may be requested from the corresponding author. The strain 'FHG000526' (TS-057) was shared based on a Material Transfer Agreement and may be requested directly from the corresponding authors of the article https://doi.org/10.1002/cbic.202100698. All the codes for each strain are available on Supplementary Data 1. Source data are provided as a Source Data file. Source data are provided in this paper.

## Code availability
The functional annotation pipeline, R scripts, and additional scripts used to create the results presented in this manuscript can be found at https://github.com/maguerreiro/Trichosporonales and https://doi.org/10.5281/zenodo.17619580.

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

## Acknowledgements

We thank Prof. Dr. Daniel Unterweger for supervising our experimental work with opportunistic human pathogens. We thank all the members of the Environmental Genomics group for support and helpful discussions. We thank the Max Planck Institute for Evolutionary Biology and the respective IT team for the computing infrastructure and technical support. This project was funded by the Deutsche Forschungsgemeinschaft (DFG) Walter Benjamin Programme (Grant No. GU 2252/1-1, Project No. 460261834) awarded to MAG. This work was in part supported by the U.S. Department of Agriculture Research Service to KB.

## Author contributions

M.A.G. conceived the study and secured funding with the support from E.H.S., A.Y. and M.N. M.A.G. performed the computational analyses and experimental assays. A.Y. and K.B. provided fungal isolates. E.H.S. provided resources and provided guidance throughout the study. M.A.G. wrote and edited the manuscript, while E.H.S. critically reviewed it. All authors contributed to discussions, reviewed and approved the final manuscript for publication.

## Funding

## Competing interests

The authors declare that they have no competing interests. Mention of trade names or commercial products in this publication is solely for the purpose of providing specific information and does not imply recommendation or endorsement by the U.S. Department of Agriculture. USDA is an equal opportunity provider and employer.
