## [Transparent Peer Review file · Nature Communications]

Genomic and physiological signatures of adaptation in pathogenic fungi

Corresponding Author: Dr Marco Guerreiro

Version 0:

Reviewer comments:

Reviewer #1

(Remarks to the Author)
Summary

In this study, the Trichosporonales order of fungi is highlighted to investigate the role of adaptive translation in the evolution of virulence in fungal pathogens. The Trichosporonales order includes both saprotrophic and opportunistic human pathogens. This species dataset was utilized to test the hypothesis that the transition of saprotrophic to pathogenicity is facilitated by adaptive translation of relevant metabolic pathways associated with each lifestyle. Codon optimization of pathways associated with saprotrophic and pathogenic lifestyles was coupled with experimental assays to test fungal fitness on substrate specific media and at temperatures relevant to human pathogens. The results are clearly communicated and convincingly indicate that adaptive translation plays a role in fungal lifestyle evolution, as well as raise important questions relating to the convergent evolution of pathogenicity.

However, we have identified a number of major and minor issues that we believe need to be addressed prior to publication.

Major Issues

Issue 1:

We have identified issues with the supplementary tRNA analysis results section. These results address interesting hypotheses of tRNA gene evolution; however, they distract from the main story. This is further confounded by the concerns we have. Our suggestion would be to exclude these results from the manuscript, or to further discuss our issues with this section (see below).

The characterisation results of tRNA gene repertoires for the fungal genomes reports instances of tRNA genes decoding histidine and aspartate being absent from 17 species (page 2). We suggest that the possibility of tRNA-scan isotype and anticodon discrepancies should also be discussed as a potential reason for these results. If tRNA-scan did identify such genes for histidine and/or aspartate, this should be shown.

The tRNA sequence variation analysis is carried out on whole tRNA gene sequences (including introns). These results are presented in the context of selection pressure, however introns are not subject to the same selection pressures as coding sequences. Therefore we suggest that justification of including introns is provided, or that only mature tRNA sequences are used in this analysis.

Issue 2:

Maximum growth rate and lag time were used as estimators for fungal fitness of 16 species. We find no evidence that species cell size has been accounted for when analyzing and drawing conclusions from the OD measurements. We suggest that if cell size has been accounted for, to state this in methods or to re-analyse this growth data.

Issue 3:

The Trichosporonales genomes used in this study include multiple isolates from the same species. We are concerned that this may have biased the results when comparing pathogenic and non-pathogenic groups. We suggest acknowledgment of the presence of multiple isolates and including a discussion of the potential bias introduced by this. Additionally, we noted

that, in Figure 2, the TE content comparison has a smaller number of genomes included. We suggest clarifying across the analyses which datasets (genomic or species level) are used and why certain genomes were omitted from this comparison.

Minor Issues

In Figure 4, the legend states that the fitness of fungal isolates is plotted in relation to translation efficiency and lifestyle. We don't observe the lifestyle information for the fungal isolates plotted on these graphs. Please define where this information is in the legend or amend the figure.

The authors propose that the phylogeny suggests human host infection arose independently multiple times (Lines 116-118). It would strengthen this argument to test this hypothesis with an ancestral reconstruction or other evolutionary test.

The terms saprotrophic (Line 47) and saprobic (53) should be defined for the broad audience of this journal.

The phrase "based on convergent tRNA gene expansion and codon optimization" (Line 265-266) is confusing. It is unclear if convergent describes the tRNA gene expansion (which does not appear to be convergent) or if it is saying that the gene expansion is convergent with the codon optimization.

The authors should elaborate on the manual curation of tRNAs (line 369). Are these tRNAs retained in the supplemental analysis?

In the statistical analyses methods (line 419), the specific type of linear models calculated are not defined.

(Remarks on code availability)

The code and underlying data appears standard and would allow a researcher to reproduce the figures and analyses conducted in the manuscript.

Reviewer #2

(Remarks to the Author)

Short assessment

In their study Guerreiro MA and coauthors apply a comparative genomics framework to explore the genetics underlying pathogenicity in the order the Trichosporonales fungi. Although mainly correlative, this work adds to the growing evidence that codon optimisation likely underpins major traits in fungi.

Short summary

In the present study, Guerreiro MA and coauthors correlate lifestyle with a set of genomic features in Trichosporonales and Tremellales fungi. Using a set of 18 opportunistic pathogens and 27 saprotrophs, they conclude that no major genetic features, i.e gene-family content, associate with lifestyle. The authors next focus on the predicted translation efficiency of different group of protein-coding genes and conclude that saprotrophic lifestyle is characterised by higher codon optimisation of genes related to carbohydrate metabolism. Finally, the authors measure the growth-rate and lag-phase of 16 species upon in vitro culture in carbohydrate or lipid-rich media and show that higher codon optimisation correlates with shorter lag-phase. In a last experiment, the authors address the link between pathogenicity and the ability to grow at elevated temperatures and conclude that growth at 37 degrees celsius is not the sole determinant of pathogenicity in the Trichosporonales.

Overall, the manuscript dives into an interesting and often overlooked aspect of genome evolution, namely codon optimisation. However, most of the results presented here are correlative and not properly integrated into their phylogenomic context.

Main comment

(1) The whole study presented here builds upon the comparative analysis of multiple fungal genomes from the Trichosporonales order, without applying the appropriate phylogenetic comparative methods. The authors conclusion that tRNA copy number / type or lifestyle has no phylogenetic signal is in contradiction with the results presented (Figures S3, S4 and S5). In addition, phylogenetic signal for lifestyle as described here was not properly tested. Globally, correlations in a phylogenomic framework should account for phylogenetic signal (i.e independent contrasts, phylogenetic generalised least squares methods).

For reference, here is a very similar study on fungi that recently investigated lifestyle transitions under the scope of codon optimisation - <https://academic.oup.com/mbe/article/39/4/msac085/6575681>

Minor comments

(1) The whole study builds on the comparison of lipid and carbohydrate-related gene pools, but no mention of how these sets were built or which genes they include. My understanding is that codon optimisation was calculated across lipid and carbohydrate-related gene pools independently of orthology assignment? Directly comparing single-copy orthogroups assigned to either functional group would help strengthen the claim that these underly adaptation.

(2) The use of different S ratios (genome or metabolism ratios) along the study is confusing and can be misleading. The S value already accounts for the genomic context so that they are directly comparable between genomes. Given the small differences highlighted in the study, it would be interesting to discuss what does a ratio of 1.12 corresponds to in terms of translation efficiency (difference of 0.09 in S values).

(3) The main result presented Figure 4 is very interesting - linking higher codon optimisation to shorter lag phase in vitro - looking at the source data (Source_growth_parameters.txt) do I understand it well that all fungi were not tested on both media (lipid and carbohydrate-rich)? Or how do lipid-optimised species perform on carbohydrate-rich media and vice-versa? Note that lifestyle does not appear in the figure 4.

(4) The last results paragraph on the growth at multiple temperatures for a subset of fungal species reads disconnected from the above. Reorganising the results section could help clarifying the rationale?

Side notes

(1) I find it hard to navigate the different species/lifestyle/S ratio datasets presented along the manuscript (i.e common pathogen, emerging pathogen, saprotroph, growth-rate, growth temperature ..), perhaps a global table summarising these different sets and what they were used for could help clarify.

(2) Regarding tRNA variation, could the authors trace back recent tRNA duplications linked to anticodon change?

(3) Many supplementary figures are not mentioned in the main results. Could grouping the supplementary results along the main results help clarify some aspects? Similarly, having all supplementary tables in one file could help navigating the data.

(4) the title and the claim in the summary that this study "demonstrate that the transition from saprotroph to opportunistic human pathogen is facilitated by adaptive translation" is overstated as most results presented are correlative and linked to a specific gene category and fungal order.

(5) Although mentioned along main results, convergence / independence / selection / adaptation is not formally addressed in the frame of this study.

Main text

(1) line 121. "convergent lifestyle-induced evolution" unclear.

(2) line 124. mentioning that some genomes were excluded here when the final set of 45 genomes was already introduced just before got me confused.

(3) line 157. why not adding supplementary results to main results?

(4) line 161. not properly tested

(5) line 180/181. "regulation" is not properly tested here

(6) line 188. selection not properly tested (note that the probability of selection acting on codon optimization can be determined using the ts.test function from the same dos Reis et al, 2004 study)

(7) line 240. higher genome / lipid / carbohydrate S?

(8) line 257. "respectively"

(9) line 292-294. "evidence that adaptive evolution shapes the composition of tRNA genes and codons to tune the expression of genes associated with distinct lifestyles" reads slightly overstated given the presented results.

(10) line 296. what is considered a codon optimised gene here?

(11) line 389. not directly inferring translational selection

(12) line 421. what does it mean? A summary table with the list of species/isolates present in each dataset / experiment could help clarify.

(Remarks on code availability)

Code

All the code and source data linked to the present work appears to be present on the provided github page. Note that I didn't try to reproduce the analysis with the code provided.

Reviewer #3

(Remarks to the Author)

The study by Guerreiro et al. investigates codon usage in fungi of the Trichosporonales order in search for genomic signatures associated with a switch from a saprotrophic to a pathogenic lifestyles. Understanding how genomes adapt to a pathogenic lifestyle is of prime conceptual and practical relevance. Regulation of virulence at the epitranscriptomic and protein translation levels is often overlooked in comparative and evolutionary genomics, so that this manuscript offers a truly original contribution to the field. The most striking finding is the discovery of a bias towards better codon optimisation in genes associated with lipid transport and metabolism relative to genes associated with carbohydrate transport and metabolism in pathogenic species. This signature associates with a faster onset of growth in vitro in lipid-rich media for pathogenic species compared to saprotrophs. The rationale for the study is straightforward and clearly explained. I found this a very interesting lead but would need additional information to fully appreciate the biological relevance of the findings.

Major comments:

1. I found that an important limitation of this report is the lack of experimental evidence that the genes and pathways identified based on codon usage are actually important for pathogenicity. If genetic manipulation is not feasible in the species under consideration, a survey of global gene expression could provide a proxy to support the involvement of lipid-related genes in the infection process.

2. Estimates for tRNA gene copy number and codon usage can be very dependent on the quality of genome assemblies and gene annotation. Some controls should be provided to ensure that genome data quality did not affect the codon usage analysis. The addition of RNAseq data might be useful to improve gene annotation if needed.

3. Another test for the biological significance of the major finding from this work would be to demonstrate that it can be used to predict yet unknown opportunistic pathogens (or strains able to initiate growth on lipids faster) in other fungal lineages. Has this approach been tested?

4. Are the observed codon usage patterns the result of drift or selection? Is the reduced codon adaptation in carbohydrate transport and metabolism genes of pathogens due to drift, following relaxed selection on codon usage (as step towards pseudogenization)? or is this rather due to an increase in translational selection on lipid transport and metabolism genes?

5. The observation that Trichonosporales species have a similar repertoire of CAZymes, secreted proteins, lipid transport and metabolism does not necessarily mean that they have the same metabolic capacity (differences in enzymes promiscuity, regulation and pathway topology may cause differences). A more systematic metabolic phenotyping of the strains or metabolomic analyses would clarify this point.

6. The strategy used in this study relies on the hypothesis that "the tRNA copy number in the genome correlates with the abundance of each tRNA family in the cell" (L370). This generally sounds like a reasonable assumption, however the context of the present study might require additional validation. Indeed, differential regulation of tRNA genes might explain why "the most frequently used synonymous codons did not consistently correspond with the most abundant tRNA gene families in the genomes" (L163). Furthermore, the relative abundance of individual tRNAs varies in response to stresses and changes in growth conditions (see for instance 10.1126/scisignal.aat6409 and 10.1093/nar/gku945). Taking into account tRNA genes regulation might reveal other pathways harboring translational selection signatures in pathogens.

Minor comments:

. L212 "Apiotrichum species classified as saprotrophic (e.g. *A. gracile* or *A. porosum*) showed a similar adaptation (Wilcoxon test $P = 1$) in both pathways and were indistinguishable from other pathogens" What seems to distinguish them is that they do not have a higher Lipid/Carbohydrate ratio than their saprotrophic relatives. This does not seem consistent with a "significantly higher translational efficiency for carbohydrate-related metabolic genes in saprotrophic species". It is not very clear to me what is special about these species and how they contribute to the hypothesis tested here.

. The experiments presented in this work indicate that there is no strict segregation between saprotrophic and opportunistic pathogen strains according to high temperature growth, this seems in contradiction with the hypothesis that "adaptation to higher temperatures may play a role in ecological niche expansion and in pathogenicity" (L259), could you please clarify?

. Is growth on lipid-rich medium affecting the expression of small secreted proteins (effectors)?

. L226 instead of "adapt", prefer "accommodate" to avoid confusion with the evolutionary term

. Title suggest the topic is about all kind of genomic signatures of adaptation across all lineages of fungi, whereas this is about one particular signature in one lineage of fungi. This does not diminish the relevance and interest of the manuscript but should be made more explicit from the title.

. Fig 2: the search for genomic signatures distinguishing saprotrophic and opportunistic pathogens might be better addressed by a PCA?

. Fig 3: Is Wilcoxon test the most adapted for comparing ratios (which might not follow a Normal distribution)?

. A N-terminal translational ramp is typical of codon-optimized genes (10.1016/j.cell.2010.03.031). Is this ramp detected in lipid and carbohydrate-related genes? more or less so between saprotrophic and pathogenic species?

. signal peptides are biased towards translationally inefficient codons (10.1016/j.bbamem.2018.09.010). Was this taken into account when comparing lipid and carbohydrate related genes?

. Fig 4: The slope of the correlation between S and lag time is steeper for growth on carbohydrates-rich medium than on lipid-rich medium, how can this be interpreted?

(Remarks on code availability)

Reviewer #4

(Remarks to the Author)

(Remarks on code availability)

Version 1:

Reviewer comments:

Reviewer #1

(Remarks to the Author)

The authors have addressed all the points raised in the original review with text edits or supplemental analysis.

The manuscript explores the hypothesis that the transition of saprotrophic to pathogenicity is facilitated by adaptive translation of relevant metabolic pathways associated with each lifestyle. This work builds on a growing field that recognizes adaptive evolution on translational efficiency as important for trait evolution. It faces several of the same challenges as other works in the field including limitations in genomic tRNA annotation. Nevertheless it contributes to our understanding of the evolution of pathogenicity.

(Remarks on code availability)

Code appears sufficient for re-analysis.

Reviewer #2

(Remarks to the Author)

Short summary

In their revised manuscript, Guerreiro et al. provide a more detailed analysis of phylogenetic signal across genome-wide metrics. They conclude that while phylogenetic relatedness explains variation in most metrics, lifestyle (pathogen vs. saprotroph) partly explains variation in codon optimisation of lipid metabolism-related genes (Table S2). They also analyse codon optimisation (S values) in single-copy orthogroups and find no overall difference between pathogens and saprotrophs. However, they report higher relative codon optimisation of metabolism-related genes in pathogens (Figure S19).

Short assessment

The additional analyses improved the manuscript. However, as presented, the results on codon optimisation are misleading and main conclusions remain speculative.

Main points

1. The codon optimisation analysis is misleading. The text suggests direct comparisons of S values (which are largely similar between lifestyles), but the main results (and conclusions) rely on S ratios, whose biological interpretation regarding translational efficiency is unclear.
2. The reported R^2 values (Pearson? Spearman?) do not account for phylogenetic non-independence (e.g. via phylogenetic independent contrasts).
3. While S is an established proxy for translational selection, the use of "translation" throughout is confusing, particularly without experimental validation. "Codon optimisation" is more precise and appropriate for the audience.

In text

line 199-209 : this part reads more related to the previous paragraph titled "tRNA composition and expansion are independent of lifestyle"

line 219 : typo "within in these pathways"

line 221-222 : "common

opportunistic pathogenic species well-defined lifestyles" reads confusing.

line 221-222 : misleading, looking at figure S16 I see that ratios are being compared, not S values. It should be clear in the text. In addition the list of genes assigned to each functional category as presented in the Table S9 is missing.

line 230-233 : misleading as here again ratios are being compared, not S values (although they are presented in Figure S17).

line 236 : presented statistics are for ratios and not S values.

line 238 : not clear in Figure S19 what the ratio is (S lipid + S carbohydrate / S genome ?)

line 242-244 : reads vague, plus S are not predictive of lifestyle but S ratios.

line 252-254 : confusing, it reads like P relates to the comparison of both pathways.

line 281-286 : clarify here that correlations are for metabolism-specific S values (and not genome-wide) in link with the growth media. Therefore "results suggest that a higher codon optimisation of metabolic pathways .."

line 309 : "is linked to a higher relative codon optimisation of the lipid metabolic pathway". tRNA gene pool not linked to lifestyle according to paragraph "tRNA composition and expansion are independent of lifestyle" and translation not experimentally tested.

line 356-357 : "the role of codon optimisation in translational speed"?

line 371 and abstract : "ecology" does not appear in results.

line 372-373 : the sentence reads redundant and confusing

(Remarks on code availability)

Reviewer #3

(Remarks to the Author)

As previously mentioned, this manuscript puts forward very interesting hypotheses on fungal pathogen evolution taking an original angle. The authors have made a commendable effort at clarifying the text and refining some analyses. I appreciated that the authors considered the use of one or multiple isolate per species (comment from reviewer 1) and tested for phylogenetic signal in several analyses (comments from reviewer 2). However, efforts at providing further experimental support has been limited. I found that the included revisions are not yet completely satisfactory and would need to be strengthened by additional information.

Important points:

1. According to the introduction, "the primary aim of [this] study was to identify genomic signatures and physiological traits associated with lifestyle and host/substrate specialization that could be used to identify potential human fungal pathogens". A key finding here is that lag phase on carbohydrates and lipids is anticorrelated with translation efficiency of the corresponding genes (Fig. 4). The extent to which this finding enables the identification of potential human pathogens remains unclear. Is there evidence for substrate specialization in the selected species and how it relates to lifestyle needs to be clarified. More thorough comparative analysis of lipid and carbohydrate metabolic pathways would be useful to determine whether orthogroups showing dramatic translation efficiency variation across species are ecologically relevant.

2. section "Translation efficiency of metabolic pathways predicts fungal lifestyle". what is the predictive value of the S ratio? In other words, what is the likelihood of correctly predicting lifestyle when shuffling S values for key metabolic pathways between species (null model)?

3. In spite of their arguments, I think the authors need to come up with a more specific and accurate title for their study (see also Reviewer 1 comment 4). First, the number of genes encoding CAZymes and other secreted is sometimes used as a signature for fungal lifestyles, whereas this parameter does not appear discriminating in the present study. Second, the lack of specificity suggests that genomic signatures have been assessed systematically, including for instance any type of gene family, although this study is focused on testing specific hypotheses. Third, the authors agree in their rebuttal letter that further functional studies will be required to establish a causal link between codon usage and fitness, but this causal relationship is implicit in the current title.

Suggestion: the authors compared codon optimization (S) across several pathways. A more systematic approach would be to search for pathways enriched among genes with the highest codon optimization. Has this approach been tested?

Minor comments and suggestions:

. L250 the S ratio used in figure 3 should be introduced in the main text

. Fig4 "Data are presented for isolates with consistent growth patterns across four independent replicates" > need to represent variability of the data along the Y axis

. The section "Saprotrophic species grow at mammal body temperatures" is not related to codon usage analysis and does add predictive power to the approach. how many times was the experiment repeated (this is not mentioned in sup fig. 21). still appears disconnected to the rest of the manuscript (reviewer 2 minor comment 4)

. "host infection arose independently multiple times": based on the ancestral state reconstruction provided, that opportunistic pathogenicity arose 3 times or that it was lost 3 times in the lineages analyses seems equally likely to me

. L186: "we observe that the tRNA composition was similar among distantly related species (Supplementary Fig. 9), suggesting convergent evolution with respect to tRNA genes." In this figure, the similarity in tRNA composition decreases with phylogenetic distance at the genus level but not beyond – this does not seem sufficient to infer some form of convergent evolution

. L188: "the most frequently used synonymous codons did not consistently correspond with the most abundant tRNA gene families in the genomes" – this seems to argue against adaptive translation at whole genome level, is there a better match between codon usage and tRNA repertoire when considering specific gene subsets?

(Remarks on code availability)

I have not tested the code but appreciate that the authors make it available. The readme file might be expanded to help readers navigate among the different files provided.

Reviewer #4

(Remarks to the Author)

(Remarks on code availability)

Version 2:

Reviewer comments:

Reviewer #2

(Remarks to the Author)

In this revised version, the authors have incorporated a phylogenetic perspective into their correlation analyses, although the methods used for this part are not described. They have also clarified the results concerning codon optimization.

However, the main conclusions still rely on comparing ratios of codon optimization values across different functional categories, and for which the biological meaning remains unclear. Since S values are directly comparable across species, there is no need to "account for interspecific variation" as stated in the text and methods. In their analysis, S values do not differ between pathogens and saprotrophs, contradicting the statement that codon optimization predicts lifestyle.

(Remarks on code availability)

Reviewer #3

(Remarks to the Author)

The authors have addressed previous comments adequately.

I still think the title of the manuscript lacks specificity and clarity but I will let the Journal Editor decide on that matter.

One point that may deserve further discussion: the authors argue that gene content is not a determinant of lifestyle in these fungal orders (L162) and later quickly elude the role of transcription (L412) with no justification. A short discussion on how the regulation of gene expression at the transcription level (in addition to protein translation) may determine lifestyle transitions, and the possible molecular mechanisms involved, seems like an important element of context to provide.

Overall, the revised manuscript provides convincing evidence for a role of adaptive translation in transition to pathogenic lifestyle in a fungal lineage, and clearly discusses the impact and limitations of the study. This is an insightful and original contribution to our understanding of fungal evolution.

Please see below my opinion on concerns on NCOMMS-24-59407B regarding the use of S ratio:

S values measure the degree of co-adaptation between the tRNA pool of a cell and codon usage in genes. They take into account both individual gene sequences (codon usage) and the cellular background (tRNA pool, generally inferred from number of copies in the genome).

S values are directly comparable across species (as said by reviewer 2) in the sense that if S is higher for gene1 in species X than in Y, it means the co-adaptation between tRNAs and codon usage for gene1 is better in X than in Y.

But a strictly identical gene sequence (same codon usage) will have different S values across species depending on the cellular background (tRNA pool). This means that the distribution range of S values in a genome varies from species to species. In other words, with a same S value, gene1 can be the best optimized gene in species X and just average relative to other genes in species Y. Its degree of co-adaptation with tRNAs will be identical strictly speaking, but the meaning for the cell functioning will be different if many other genes outcompete it for tRNAs recruitment (species Y) or not (species X). This is probably what the authors call "interspecific variation".

For this reason, I think the ratio approach of the authors has merits and offers biological insights by showing with a single metric which of two pathways has better co-evolved with the tRNA pool across species. But I agree with reviewer 2 that they need to be more specific so that their conclusions are not confusing. Typically, if conclusions are based on S1/S2 ratio comparisons, then the "relative codon optimization of pathway 1 and 2 predicts lifestyle" and not simply "codon optimization".

In short, my recommendations would be as follows:

- 1) - More clearly present and comment S values for each pathway in each species, as well as the distribution of S values in all the genomes analyzed (as supplementary material) to document the effect of codon usage and tRNA pool variation on the S metrics, so that the choice of ratios is clearly justified
- 2) - Another approach to compare individual genes across species would be to quantile normalize S values across genomes. Individual normalized S values would then be directly comparable across species (top S values will be equal in all species). Personally, I had favored normalization as it makes the whole S value distributions comparable and does not

rely on one reference gene/pathway as the ratio does. I would suggest the authors to try this approach and verify that they reach the same conclusions as with ratios.

3) - Invite authors to carefully revise their manuscript to make sure interpretations and conclusions are strictly accurate, that the analysis strategy is properly and fully justified. A short paragraph dedicated to the interpretation of the S ratio/normalized S/raw S values and their meaning in the context of this study might be useful to add to the discussion section.

(Remarks on code availability)

I have not tested the code but appreciate that the authors make it available

Point-to-point response to reviewers' comments

Reviewer #1 (Remarks to the Author):

Summary

In this study, the Trichosporonales order of fungi is highlighted to investigate the role of adaptive translation in the evolution of virulence in fungal pathogens. The Trichosporonales order includes both saprotrophic and opportunistic human pathogens. This species dataset was utilized to test the hypothesis that the transition of saprotrophic to pathogenicity is facilitated by adaptive translation of relevant metabolic pathways associated with each lifestyle. Codon optimization of pathways associated with saprotrophic and pathogenic lifestyles was coupled with experimental assays to test fungal fitness on substrate specific media and at temperatures relevant to human pathogens. The results are clearly communicated and convincingly indicate that adaptive translation plays a role in fungal lifestyle evolution, as well as raise important questions relating to the convergent evolution of pathogenicity.

However, we have identified a number of major and minor issues that we believe need to be addressed prior to publication.

Major Issues

Issue 1:

We have identified issues with the supplementary tRNA analysis results section. These results address interesting hypotheses of tRNA gene evolution; however, they distract from the main story. This is further confounded by the concerns we have. Our suggestion would be to exclude these results from the manuscript, or to further discuss our issues with this section (see below).

Response: We appreciate the remarks on the supplementary results. Although we recognize that these results might be distracting, we believe that the tRNA evolution is an important side aspect to support our main conclusions. Therefore, we would like to keep them as supplementary information. The evolution of tRNA gene composition, expansion and sequence is a fundamental component of adaptive translation, and these results offer additional support for the observed patterns related to lifestyles. To ensure that the manuscript maintains its focus and is within space limitations, we decided to keep these results as supplementary. However, we are open to include some of them in the main text, if considered essential.

The characterisation results of tRNA gene repertoires for the fungal genomes reports instances of tRNA genes decoding histidine and aspartate being absent from 17 species (page 2). We suggest that the possibility of tRNA-scan isotype and anticodon discrepancies should also be discussed as a

potential reason for these results. If tRNA-scan did identify such genes for histidine and/or aspartate, this should be shown.

Response: During the tRNA annotation with tRNAscan, some tRNA genes were indeed classified as potential pseudogenes and we excluded these from further analyses. However, some of these pseudogenes were potentially decoding histidine and aspartate. It is important to note that there was no bias towards these tRNA gene families decoding histidine and aspartate. Additionally, we excluded tRNA genes with a covariance score <30, as recommended by literature. While this approach increases the confidence of the gene prediction and minimizes false positives, it may also contribute to the observed absence of certain tRNA genes. We now include a supplementary table (Supplementary Table 7) reporting the number of pseudogenes that were excluded, per genome and address this issue in the Supplementary Results and Supplementary Discussion.

In the respective section of the Supplementary Results, we now include the following underlined sentence:

“However, tRNAs genes decoding histidine (tRNA^{His}) were in low abundance overall and even absent in 15 genomes (Ap. domesticum, Ap. montevidense, Ap. porosum, Cutaneotrichosporon arboriformis, Cu. curvatus, Cu. cyanovorans, Cu. dermatis, P. fragicola, Trichosporon asahii, and Tr. inkin), while the aspartate-encoding tRNA (tRNA^{Asp}) was absent in Pascua and Takashimella spp. Some of these genes were, however, detected as pseudogenes during tRNA prediction (Supplementary Table 7).”

In the Supplementary Discussion we added the following underlined sentence:

“Previous studies have reported that codon usage and codon abundance in protein-coding genes do not always correspond to the most abundant associated tRNA gene, and some tRNAs may even remain undetected in the genome^{2,4}. It has been suggested that in fungi, tRNA wobbling of the third base position might enable a tRNA to recognize several unspecific synonymous codons and thereby allow the decoding of these amino acids^{2,4,5}. The absence of some tRNA could also be due to technical limitations of predictions tools, such as classification of tRNAs as pseudogenised genes (Supplementary Table 7) or incorrect assignment of the isotype.”

The tRNA sequence variation analysis is carried out on whole tRNA gene sequences (including introns). These results are presented in the context of selection pressure, however introns are not subject to the same selection pressures as coding sequences. Therefore we suggest that justification of including introns is provided, or that only mature tRNA sequences are used in this analysis.

Response: Thank you for pointing out this very important point. We completely acknowledge that the selection pressure is different between introns and exons of tRNA genes. In this section, our aim was to analyse the overall evolutionary dynamics of the tRNA genes, including introns. This approach allowed us to address gene duplication and expansion events (e.g. Supplementary Figure 6), for which diversification of the gene sequence might be more pronounced within the intronic regions. Since our focus was not limited to the functional regions, we consider that including introns provide valuable information in an evolutionary context. We added an explanation to the text.

We now include in the Materials and Methods the following underlined changes for clarity:

“To explore the overall evolutionary dynamics of tRNA genes, including gene duplications and expansion events, we analyzed the full-length tRNA gene sequences, including introns. To estimate sequence variation, the complete tRNA gene sequences were obtained for each genome and aligned for each anticodon type using MAFFT v6.864b ⁴⁶”

Issue 2:

Maximum growth rate and lag time were used as estimators for fungal fitness of 16 species. We find no evidence that species cell size has been accounted for when analyzing and drawing conclusions from the OD measurements. We suggest that if cell size has been accounted for, to state this in methods or to re-analyse this growth data.

Response: To account for potential biases introduced by cell size, we counted CFUs among isolates for the same initial absorbance that was used in these assays ($OD_{600nm} = 2.5 \times 10^{-4}$). We observed that the initial number of CFUs (approx. 100 – 1500 CFUs) across isolates did not correlate with any growth parameter and could not explain the growth rate or the lag time. Given this, we consider that the cell size does not play a role in these particular experiments. We added additional information in the materials and methods section to address this issue. Additionally, we now include a supplementary table (Supplementary Table 12) with the CFU counts per isolate at the initial OD of the growth experiments ($OD_{600nm} = 2.5 \times 10^{-4}$).

Issue 3:

The Trichosporonales genomes used in this study include multiple isolates from the same species. We are concerned that this may have biased the results when comparing pathogenic and non-pathogenic groups. We suggest acknowledgment of the presence of multiple isolates and including a discussion of the potential bias introduced by this.

Response: Thank you for bringing up this important observation. We indeed performed all analyses considering only one isolate per species. The results for single isolates were congruent with the ones obtained with multiple isolates per species (e.g. *Trichosporon asahii* or *Cutaneotrichosporon oleaginosus*). For some of the analyses (e.g. Fig. 2) only one isolate per species was considered in the analyses. We now provide an additional supplementary figure for Fig. 3 (Supplementary Figure 18), where we include only one isolate per species. We find relevant to show on Fig. 3 that, while even considering interspecies variation, our results are consistent among individuals of the same species and not specific to one genome. In the Supplementary Table 1 we indicate which genomes were considered as representatives for the species. Additionally, we now clarify on the legend of Fig. 3 that the analysis considers multiple isolates per species by adding the following information:

“In order to consider interspecific variation, the analysis considers several genomes for some species. The results considering 1 genome per species are consistent with the ones presented and are shown in Supplementary Figure 18.”

Additionally, we noted that, in Figure 2, the TE content comparison has a smaller number of genomes included. We suggest clarifying across the analyses which datasets (genomic or species level) are used and why certain genomes were omitted from this comparison.

Response: Thank you for noticing that we missed to clarify this detail in the text. During TE identification, the pipeline we used (REPET3) was unable to identify any TEs in some genomes. We believe this might be due to the sequencing technologies used or assembly methods in the respective original studies. Therefore, these genomes are omitted in this particular panel. We now added, in the methods section, the following underlined sentences for clarity:

“Transposable elements were predicted by analyzing each genome assembly with REPET3 pipeline^{59,60}, and according to previous studies⁶¹. Possibly due to sequencing technologies or assembly methods, the REPET3 approach was unable to predict TEs for some genomes. Therefore, the analyses regarding TEs consider a lower number of species.”

Minor Issues

In Figure 4, the legend states that the fitness of fungal isolates is plotted in relation to translation efficiency and lifestyle. We don't observe the lifestyle information for the fungal isolates plotted on these graphs. Please define where this information is in the legend or amend the figure.

Response: Indeed, there is no lifestyle depicted in the figure. We corrected the legend.

The authors propose that the phylogeny suggests human host infection arose independently multiple times (Lines 116-118). It would strengthen this argument to test this hypothesis with an ancestral reconstruction or other evolutionary test.

Response: We appreciate the suggestion. We now provide an ancestral reconstruction and additional phylogenetic tests that account for the influence of phylogenetic structure and lifestyle on our analyses. The ancestral reconstruction indicates that the pathogenic lifestyle emerged in the Trichosporonales independently in the *Apiotrichum* genus and in the clade containing the *Trichosporon* and *Cutaneotrichosporon* genera. We also noted that the ancestral reconstruction indicates a second lifestyle transition event in the pathogenic *Cutaneotrichosporon* genus to an exclusively saprotrophic lifestyle.

This reconstruction supports our main conclusion that lifestyle transitions among this group of fungi are dynamic.

We now provide this reconstruction in the supplementary figures.

The terms saprotrophic (Line 47) and saprobic (53) should be defined for the broad audience of this journal.

Response: We use these two terms as synonyms. For consistency and clarity, we replaced the saprobic term with saprotrophic throughout the text and added a definition in the introduction.

“Some saprotrophic species (i.e. which acquire nutrients from dead organic matter) can become opportunistic human pathogens...”

The phrase “based on convergent tRNA gene expansion and codon optimization” (Line 265-266) is confusing. It is unclear if convergent describes the tRNA gene expansion (which does not appear to be convergent) or if it is saying that the gene expansion is convergent with the codon optimization.

Response: We recognize that this the sentence could be confusing and corrected it.

The authors should elaborate on the manual curation of tRNAs (line 369). Are these tRNAs retained in the supplemental analysis?

Response: During manual curation, we manually verified a subset of tRNA genes secondary structures. All of the gene structures were as expected. We performed this curation step in order to verify that excessive copy numbers were not due to automatic gene annotation errors. This was performed based on the quality-filtered genes, filtered based on covariance score and pseudogene classification. Therefore, all the curated sequences and all the quality-filtered genes were retained.

In the statistical analyses methods (line 419), the specific type of linear models calculated are not defined.

Response: We now clarify this in the methods.

“Linear regression models, R values, and *P* values were calculated with the *ggpmisc* R package⁷³”

Reviewer #1 (Remarks on code availability):

The code and underlying data appears standard and would allow a researcher to reproduce the figures and analyses conducted in the manuscript.

Response: We appreciate the time and effort in reviewing our codes and data. Our code and data in the GitHub repository were updated to reflect the modifications on our analyses, including the new analyses.

Reviewer #2 (Remarks to the Author):

Short assessment

In their study Guerreiro MA and coauthors apply a comparative genomics framework to explore the genetics underlying pathogenicity in the order the Trichosporonales fungi. Although mainly correlative, this work adds to the growing evidence that codon optimisation likely underpins major traits in fungi.

Short summary

In the present study, Guerreiro MA and coauthors correlate lifestyle with a set of genomic features in Trichosporonales and Tremellales fungi. Using a set of 18 opportunistic pathogens and 27 saprotrophs, they conclude that no major genetic features, i.e gene-family content, associate with lifestyle. The authors next focus on the predicted translation efficiency of different group of protein-coding genes and conclude that saprotrophic lifestyle is characterised by higher codon optimisation of genes related to carbohydrate metabolism. Finally, the authors measure the growth-rate and lag-phase of 16 species upon in vitro culture in carbohydrate or lipid-rich media and show that higher codon optimisation correlates with shorter lag-phase. In a last experiment, the authors address the link between pathogenicity and the ability to grow at elevated temperatures and conclude that growth at 37 degrees celsius is not the sole determinant of pathogenicity in the Trichosporonales.

Overall, the manuscript dives into an interesting and often overlooked aspect of genome evolution, namely codon optimisation. However, most of the results presented here are correlative and not properly integrated into their phylogenomic context.

Response: We appreciate the recognition of our study and all the comments provided. We now provide more clear evidence regarding the phylogenomic signal in several of our analyses.

Main comment

(1) The whole study presented here builds upon the comparative analysis of multiple fungal genomes from the Trichosporonales order, without applying the appropriate phylogenetic comparative methods.

Response: We appreciate the reviewer's criticism regarding the phylogenetic signal on our comparative analyses. We now address the question by testing for the phylogenetic signal in our main analyses, following the approach outlined by Mesny and Vannier (2020), which was previously applied in the study by Hill et al. 2022, to test for the influence of both phylogeny and lifestyle on our results. For this, the phylogenetic structure was decomposed into principal components and tested together with lifestyle to assess their effects on different genomic datasets.

In these results, we observe that the variance in genome composition and structure is explained mainly by phylogeny and not by lifestyle. In addition, the variance in the parameter "S" of genes related to carbohydrate transport and metabolism is also only explained by the phylogeny. However, the parameter "lifestyle" explains more than the phylogeny regarding the variance in the S of genes

related to lipid transport and metabolism. Additionally, the most striking observation is on the ratio of the S of these two pathways, where the phylogeny does not explain the variance of S, but rather the lifestyle explains the variance in S with a coefficient of determination (R^2) of 0.69.

In summary, we consider that lifestyle itself is not a phylogenetically inherited trait in our dataset, since both pathogenic and saprotrophic species are present in multiple clades, further indicating that lifestyle is not due to phylogenetic relatedness. We appreciate the recommendation by the reviewer and we now include this table in the supplements to support our claims.

Dataset	Formula	Effect	Df	R2	F	p
All orthologous genes count	All OGs ~ PC1 + PC2 + lifestyle	PC1	1	0.16	9.70	1e-04
		PC2	1	0.10	6.20	1e-04
		lifestyle	1	0.05	3.10	5e-04
All genome metrics	Stats ~ PC1 + PC2 + lifestyle	PC1	1	0.21	13.33	0.0001
		PC2	1	0.12	7.80	0.0031
		lifestyle	1	0.02	0.97	0.3456
Metrics – Genome size	size ~ PC1 + PC2 + lifestyle	PC1	1	0.21	14.14	0.0004
		PC2	1	0.16	10.96	0.0006
		lifestyle	1	0.02	1.38	0.2422
Metrics – GC content	GC ~ PC1 + PC2 + lifestyle	PC1	1	0.52	60.21	0.0001
		PC2	1	0.13	15.06	0.0002
		lifestyle	1	0.02	2.85	0.0894
Metrics – tRNA genes count	tRNAs ~ PC1 + PC2 + lifestyle	PC1	1	0.26	17.56	0.0001
		PC2	1	0.13	8.89	0.0003
		lifestyle	1	0.04	2.44	0.0712
Metrics – Repeat content	repeats ~ PC1 + PC2 + lifestyle	PC1	1	0.10	4.48	0.0240
		PC2	1	0.30	1.42	0.2434
		lifestyle	1	0.00	0.13	0.9034
Metrics – Proteins count	proteins ~ PC1 + PC2 + lifestyle	PC1	1	0.18	9.67	0.0024
		PC2	1	0.00	0.16	0.7944
		lifestyle	1	0.04	2.08	0.1447
Metrics – Secreted proteins count	Sec-proteins ~ PC1 + PC2 + lifestyle	PC1	1	0.30	17.76	0.0001
		PC2	1	0.01	0.79	0.4272
		lifestyle	1	0.01	0.76	0.4293
Metrics – CAZymes count	CAZymes ~ PC1 + PC2 + lifestyle	PC1	1	0.16	9.97	0.0014
		PC2	1	0.11	6.88	0.0080
		lifestyle	1	0.02	1.07	0.3037
Metrics – Secreted CAZymes count	Sec-CAZymes ~ PC1 + PC2 + lifestyle	PC1	1	0.20	11.69	0.0006
		PC2	1	0.12	7.05	0.0070
		lifestyle	1	0.01	0.70	0.4547
Metrics – Carbohydrate genes count	carbos ~ PC1 + PC2 + lifestyle	PC1	1	0.12	6.65	0.0101
		PC2	1	0.03	1.43	0.2307
		lifestyle	1	0.05	2.65	0.1005
Metrics – Lipid genes count	lipids ~ PC1 + PC2 + lifestyle	PC1	1	0.74	144.72	0.0001
		PC2	1	0.04	7.47	0.0052
		lifestyle	1	0.01	2.83	0.0864
tRNA gene counts by anticodon	tRNA anticodon ~ PC1 + PC2 + lifestyle	PC1	1	0.24	15.94	0.0001
		PC2	1	0.12	7.75	0.0001
		lifestyle	1	0.03	2.25	0.0674
S carbohydrate transport and metabolism for Cutaneotrichosporon , Trichosporon , and Cryptococcus	S carbos ~ PC1 + PC2 + lifestyle	PC1	1	0.59	31.16	0.0001
		PC2	1	0.40	20.98	0.0001
		lifestyle	1	0.06	3.18	0.0809

S lipid transport and metabolism for Cutaneotrichosporon , Trichosporon , and Cryptococcus	S lipids ~ PC1 + PC2 + lifestyle	PC1	1	0.24	17.22	0.0006
		PC2	1	0.14	9.92	0.0049
		lifestyle	1	0.59	41.64	0.0001
S carbohydrate:lipid ratio for Cutaneotrichosporon , Trichosporon , and Cryptococcus	S ratio ~ PC1 + PC2 + lifestyle	PC1	1	0.01	0.43	0.5485
		PC2	1	0.01	0.52	0.5069
		lifestyle	1	0.69	40.96	0.0001

The authors conclusion that tRNA copy number / type or lifestyle has no phylogenetic signal is in contradiction with the results presented (Figures S3, S4 and S5).

In addition, phylogenetic signal for lifestyle as described here was not properly tested.

Response: We recognize the issues with our conclusions based on these analyses. We corrected our statements related to the tRNA evolution in relation to lifestyle and phylogeny, accordingly. Additionally, we now test for phylogenetic signal in our analyses. These indeed show that the tRNA copy number and type are shaped only by phylogenetic structure. After including new analyses where we correct for the phylogenetic signal, the main message of our study remains the same, that codon optimization and translation efficiency are linked to lifestyle in this group of fungi.

Globally, correlations in a phylogenomic framework should account for phylogenetic signal (i.e independent contrasts, phylogenetic generalised least squares methods).

For reference, here is a very similar study on fungi that recently investigated lifestyle transitions under the scope of codon optimisation - <https://academic.oup.com/mbe/article/39/4/msac085/6575681>

Response: We appreciate the recommendation and we followed the approaches used in the study to account for phylogenetic signal in our analyses (please see above).

Minor comments

(1) The whole study builds on the comparison of lipid and carbohydrate-related gene pools, but no mention of how these sets were built or which genes they include. My understanding is that codon optimisation was calculated across lipid and carbohydrate-related gene pools independently of orthology assignment? Directly comparing single-copy orthogroups assigned to either functional group would help strengthen the claim that these underly adaptation.

Response: Thank you for pointing this out. The analyses related to the pathways were based on the functional annotations of each predicted gene by eggNOG, namely the narrowest orthologous groups. Indeed, the codon optimisation was calculated independently of orthology. We now clarify this in the

materials and methods section. We also include an additional table with the eggNOG annotation for each gene considered in these datasets.

The direct comparison of single-copy orthogroups present in all tested species with well-defined lifestyles (*Cryptococcus*, *Cutaneotrichosporon* and *Trichosporon*) shows the trend that opportunistic pathogens have a higher codon optimization in genes related to lipid metabolism than the saprotrophic species ($P = 0.046$). Due to the reduced dataset, the P-value remains significant but close to 0.05, however the patterns are still observable.

We now include these results in the supplementary figures.

(2) The use of different S ratios (genome or metabolism ratios) along the study is confusing and can be misleading. The S value already accounts for the genomic context so that they are directly comparable between genomes. Given the small differences highlighted in the study, it would be interesting to discuss what does a ratio of 1.12 corresponds to in terms of translation efficiency (difference of 0.09 in S values).

Response: We understand that the usage of different ratios might be confusing, but by normalizing these values, we are able to compare the relative codon optimization across species. We agree that S accounts for the genomic context, however, we also consider that accounting for species-level differences in codon usage is necessary when comparing different species. Therefore, we considered ratios as a normalization step to remove species-specific biases in order to compare whether a particular subset of genes is more or less optimized relatively to the rest of the genome, rather than focusing exclusively on absolute S values. However, we also considered the absolute S values, and we provide these analyses on Supplementary Figure 17, where both absolute and normalized S values are compared for the different pathways.

We are happy to provide additional discussion; however, we were not able to find the context of the 1.12 ratio regarding the translation efficiency or the difference of 0.09 in the S values mentioned by the reviewer.

(3) The main result presented Figure 4 is very interesting - linking higher codon optimisation to shorter lag phase in vitro - looking at the source data (Source_growth_parameters.txt) do I understand it well that all fungi were not tested on both media (lipid and carbohydrate-rich)? Or how do lipid-optimised species perform on carbohydrate-rich media and vice-versa? Note that lifestyle does not appear in the figure 4.

Response: For these analyses, we performed the experiments with 4 replicates per isolate. When analysing the growth curves for each isolate, we excluded the data from the isolate from further analyses when replicates were not consistent or when the isolate showed an abnormal growth. Due to some degree of dimorphism of some isolates under certain conditions, we were unable to collect consistent data after several experiments for these isolates. Therefore, the dataset is incomplete. We note the challenge of standardizing experimental procedures across a large diversity of fungal species. The isolates for which the data is presented showed a yeast-like growth and the growth among replicates were comparable.

We corrected the legend of figure 4 to make our procedures and handling of replicates clearer.

(4) The last results paragraph on the growth at multiple temperatures for a subset of fungal species reads disconnected from the above. Reorganising the results section could help clarifying the rationale?

Response: We acknowledge that this paragraph might seem disconnected from the above, however, we believe these last 2 sections follow a logical progression of our results.

We suggest in the last sentence of the translation efficiency section that the translation efficiency contributes to fitness advantage (adaptation speed). The temperature growth section follows naturally by addressing the growth temperature, which is especially relevant for transitioning from saprotrophic to pathogenic lifestyles.

We acknowledge that some rephrasing and sentence restructuring could improve the clarity of the results, however we believe the current paragraph structure provides coherence. However, we are open for suggestions to rephrase or restructure segments of the text which might be unclear or missing context to the reviewer.

Side notes

(1) I find it hard to navigate the different species/lifestyle/S ratio datasets presented along the manuscript (i.e common pathogen, emerging pathogen, saprotroph, growth-rate, growth temperature ..), perhaps a global table summarising these different sets and what they were used for could help clarify.

Response: We appreciate the feedback of the reviewer and we will take this into account to refine the presentation of our results. We understand that different concepts throughout the study might be confusing, however, we believe that the current organization of the manuscript provides sufficient context for each dataset and metric. Each dataset (e.g., common pathogen, emerging pathogen, saprotroph, growth rate, growth temperature) is introduced in the respective section, ensuring that their purpose to the analysis and meaning are clear. Given this structure, we do not find it necessary to include a global summary table. However, we will ensure that these concepts and differences are clear within each section and we are adding brief clarifications where needed to improve clarity.

(2) Regarding tRNA variation, could the authors trace back recent tRNA duplications linked to anticodon change?

Response: This is indeed an interesting question. While in this study we did not specifically investigate anticodon changes, we were able to identify tRNA gene sequences differing only by 1 bp in the anticodon, however these were encoding for the same amino acid (e.g. tRNA-AGC and tRNA-CGC encoding Ala). In these cases, the intronic regions were very different, implying that this might not be a recent event. We additionally identified tRNAs encoding different amino acids that were rather similar in sequence, but already differed in some positions in the tRNA body sequence. We are not able to say if this is indeed a duplication followed by a few mutations leading to a codon change. Due to the observed sequence diversity, a fast mutation rate is a possible scenario, which is also observed in other publications (e.g. Yona et al. 2013). To properly address this question, we are currently planning a future study on anticodon shifts in tRNA genes, where we could address the evolution of codon recognition and translational adaptation.

(3) Many supplementary figures are not mentioned in the main results. Could grouping the supplementary results along the main results help clarify some aspects? Similarly, having all supplementary tables in one file could help navigating the data.

Response: We have merged all the supplementary tables into a single Excel file. Additionally, we revised the text and we now mention all the supplementary information in the main text.

(4) the title and the claim in the summary that this study "demonstrate that the transition from saprotroph to opportunistic human pathogen is facilitated by adaptive translation" is overstated as most results presented are correlative and linked to a specific gene category and fungal order

Response: We think our experimental and computational analyses provide evidence for a hitherto poorly considered mechanism of fungal niche adaptation, namely by modification of the translation efficiency. In the revised version of the manuscript, we include new analyses. Hereby we revisit several of our analyses to account for a putative phylogenetic signal in the results. These results clearly demonstrate that phylogenetic relatedness does not explain the observed distribution of fungal

lifestyles. Rather our analyses support dynamic transitions of lifestyles among fungal species, which to some extent can be explained by translation efficiency of key pathways.

We are aware that future functional studies focusing on specific genes or certain species, would be of great interest to further understanding niche adaptation. However, the scope of this study was the comparative analyses of a large number of species.

(5) Although mentioned along main results, convergence / independence / selection / adaptation is not formally addressed in the frame of this study.

Response: We believe that our study reveals adaptive evolution occurring at the level of translation efficiency. We have discussed the role of natural selection in shaping translation efficiency of different groups of genes (adaptation to different substrates), and how to see this in the light of convergence. If the reviewer feels that these terms should be further emphasized and explained, we will be happy to follow further advice to specific elements of the manuscript.

Main text

(1) line 121. "convergent lifestyle-induced evolution" unclear.

Response: We rephrased this sentence. It now reads:

“To identify genomic signatures associated with lifestyle and with convergent evolution driven by similar ecological environments, we characterized and compared the genomic structure and composition between opportunistic pathogens and saprotrophic species in the order Trichosporonales.”

(2) line 124. mentioning that some genomes were excluded here when the final set of 45 genomes was already introduced just before got me confused.

Response: We restructured this part of the results for clarity. We now present the total number of genomes and the criteria to exclude some of these before mentioning the final set of 45 genomes.

(3) line 157. why not adding supplementary results to main results?

Response: We now add additional supplementary results to the main text. We acknowledge that these results provide additional relevant details, however, to maintain clarity and focus on the main findings, as suggested by Reviewer 1 (see above), we have decided to keep most of these results in the supplementary materials.

(4) line 161. not properly tested

Response: This sentence has been edited. We now include additional tests addressing the phylogenetic context and corrected some of our interpretations throughout the text.

(5) line 180/181. "regulation" is not properly tested here

Response: We replaced "regulation" with "influence".

(6) line 188. selection not properly tested (note that the probability of selection acting on codon optimization can be determined using the ts.test function from the same dos Reis et al, 2004 study)

Response: We appreciate mentioning this important point. We now provide the results from the statistical test for the determination of *S*. Instead of using a Monte Carlo test of permutation, we performed Pearson and Spearman correlation tests, following the same principles as the *ts.test* function. We observed that all the *S* values used in our study were significant (*P*-value < 0.05) by both correlation tests. For consistency, since the *S* values are calculated based on Pearson correlations, we now provide the respective Pearson *P*-values in the Supplementary Tables 9 and 10.

(7) line 240. higher genome / lipid / carbohydrate S?

Response: We rephrased this sentence. It now reads:

"However, we observed a significant correlation ($R^2 > 0.6$, *P*-value < 0.013) between lag time and *S* values (Fig. 4B), indicating that isolates with higher *S* of the corresponding metabolic pathway displayed shorter lag phases."

(8) line 257. "respectively"

Response: We corrected this.

(9) line 292-294. "evidence that adaptive evolution shapes the composition of tRNA genes and codons to tune the expression of genes associated with distinct lifestyles" reads slightly overstated given the presented results.

Response: We rephrased this sentence. It now reads:

“Based on our comparative genome analysis and our *in-vitro* fitness assays, our results indicate that adaptive evolution may shape the composition of tRNA genes and codons to tune the expression of genes associated with distinct lifestyles ²⁴.”

(10) line 296. what is considered a codon optimised gene here?

Response: We edited the sentence and now clarify that this is “optimal codon usage”.

(11) line 389. not directly inferring translational selection

Response: We rephrased this sentence. It now reads:

“To assess for potential translational selection, the tRNA adaptation index (tAI) and codon optimization index (S) were calculated with the tAI v0.2.1 R package ²⁵. Here, we consider tAI as a proxy for translation efficiency and S as an estimate of the strength of translational selection, respectively.”

(12) line 421. what does it mean? A summary table with the list of species/isolates present in each dataset / experiment could help clarify.

Response: We now clarify in the figure legends which isolates are being considered. Regarding the genomic data, we indicate in Supplementary Table 1 which individuals were considered as representatives of each species.

Reviewer #2 (Remarks on code availability):

Code

All the code and source data linked to the present work appears to be present on the provided github page. Note that I didn't try to reproduce the analysis with the code provided.

Response: We appreciate the time and effort in reviewing our codes and data. Our code and data in the GitHub repository were updated to reflect the modifications on our analyses, including the new analyses.

Reviewer #3 (Remarks to the Author):

The study by Guerreiro et al. investigates codon usage in fungi of the Trichosporonales order in search for genomic signatures associated with a switch from a saprotrophic to a pathogenic lifestyles. Understanding how genomes adapt to a pathogenic lifestyle is of prime conceptual and practical relevance. Regulation of virulence at the epitranscriptomic and protein translation levels is often overlooked in comparative and evolutionary genomics, so that this manuscript offers a truly original contribution to the field. The most striking finding is the discovery of a bias towards better codon optimisation in genes associated with lipid transport and metabolism relative to genes associated with carbohydrate transport and metabolism in pathogenic species. This signature associates with a faster onset of growth in vitro in lipid-rich media for pathogenic species compared to saprotrophs. The rationale for the study is straightforward and clearly explained. I found this a very interesting lead but would need additional information to fully appreciate the biological relevance of the findings.

Major comments:

1. I found that an important limitation of this report is the lack of experimental evidence that the genes and pathways identified based on codon usage are actually important for pathogenicity. If genetic manipulation is not feasible in the species under consideration, a survey of global gene expression could provide a proxy to support the involvement of lipid-related genes in the infection process.

Response: The experimental validation of gene function is currently out of the scope of our study. However, gene expression analyses of *Cryptococcus neoformans* (also analysed in our study) during infection show that genes involved in the metabolism of carbohydrates and lipids are reported to be highly expressed and also enriched in human cerebrospinal fluid (<https://doi.org/10.1128/mBio.02313-21>). Although we did not analyse the gene expression in our datasets, the above-mentioned study supports our hypotheses that these gene sets are involved in pathogenesis and that codon optimization in these pathways are linked to the pathogenic lifestyle.

We added the following sentence to the discussion to reflect this limitation on our study:

“Although we did not analyze gene expression in our study, previous studies on *Cryptococcus deneoformans* show that genes involved in carbohydrate and lipid metabolism are highly expressed during infection and enriched in human cerebrospinal fluids³⁷, supporting our hypothesis that these gene sets are important for pathogenesis.”

2. Estimates for tRNA gene copy number and codon usage can be very dependent on the quality of genome assemblies and gene annotation. Some controls should be provided to ensure that genome data quality did not affect the codon usage analysis. The addition of RNAseq data might be useful to improve gene annotation if needed.

Response: We recognize the concern regarding the impact of the genome quality. To prevent and mitigate any biases, we processed all genomes with the same pipelines and settings and used only genomes with at least 85% completeness on their predicted proteome. Additionally, we have performed manual curation of some of the predicted tRNA genes to validate the predictions. We also

observe that the completeness of a genome does not correlate with the corresponding tRNA gene copy number, by applying a linear regression model ($R^2 = 0.08$, $P > 0.05$) and correlation tests with correction (adjusted P) for multiple correlations (Pearson P.adj > 0.05 ; Spearman P.adj > 0.05 ; Kendall P.adj > 0.05). We also verified that our results (131 tRNA genes, 8 tRNA pseudogenes, Supplementary Tables 6 and 7) are in accordance with what is known for some of the species, namely *Cryptococcus deneoformans* (131 tRNA genes, 8 tRNA pseudogenes according to https://gtrnadb.ucsc.edu/GtRNADB2/genomes/eukaryota/Cryp_neof_var_neoformans_JEC21/). We are confident on our tRNA gene predictions and we believe that adding RNAseq data for improved gene annotation is out of the scope of the current study.

3. Another test for the biological significance of the major finding from this work would be to demonstrate that it can be used to predict yet unknown opportunistic pathogens (or strains able to initiate growth on lipids faster) in other fungal lineages. Has this approach been tested?

Response: That is a great question and we are currently planning to explore it in detail in a follow-up study on this topic.

4. Are the observed codon usage patterns the result of drift or selection? Is the reduced codon adaptation in carbohydrate transport and metabolism genes of pathogens due to drift, following relaxed selection on codon usage (as step towards pseudogenization)? or is this rather due to an increase in translational selection on lipid transport and metabolism genes?

Response: This is an extremely important and an intriguing question, however it is challenging to distinguish between genetic drift and relaxed selection acting on codon usage with the current datasets available. What we can currently say, is that our results indicate that the codon optimization on the lipid pathway is highly influenced by lifestyle. It is possible that the increased codon optimization in the lipid pathway in pathogenic species might reflect translational selection favouring translation of these proteins rather than relaxed selection, since both pathways are crucial for survival.

5. The observation that Trichonosporales species have a similar repertoire of CAZymes, secreted proteins, lipid transport and metabolism does not necessarily means that they have the same metabolic capacity (differences in enzymes promiscuity, regulation and pathway topology may cause differences). A more systematic metabolic phenotyping of the strains or metabolomic analyses would clarify this point.

Response: We agree that having a similar gene repertoire does not necessarily imply identical metabolic capacity. Although our study focuses on genome evolution, a comprehensive metabolic phenotyping combined with genomic data could help clarify their metabolic capacity. This is an important direction for future work that we are currently also considering to better understand the metabolic diversity and pathogenesis within the Trichosporonales order.

6. The strategy used in this study relies on the hypothesis that "the tRNA copy number in the genome correlates with the abundance of each tRNA family in the cell" (L370). This generally sounds like a reasonable assumption, however the context of the present study might require additional validation. Indeed, differential regulation of tRNA genes might explain why "the most frequently used synonymous codons did not consistently correspond with the most abundant tRNA gene families in the genomes" (L163). Furthermore, the relative abundance of individual tRNAs varies in response to stresses and changes in growth conditions (see for instance 10.1126/scisignal.aat6409 and 10.1093/nar/gku945). Taking into account tRNA genes regulation might reveal other pathways harboring translational selection signatures in pathogens.

Response: We appreciate the insightful comment. The idea of considering tRNA gene regulation during infection is extremely intriguing and might indeed help us find additional translational selection signatures. The relative abundance of individual tRNA genes varies in responses to stress and this is often overlooked in fungal biology. Our study considers the genomic tRNA copy number as a proxy for the tRNA pool in the cell, however, we acknowledge that the optimum conditions might differ for each isolate, and due to different levels of stress, the tRNA abundance might be different from the theoretical. It will be important to determine how stress conditions affect the tRNA pool in the cell to further understand the role of translational selection in different lifestyles and during challenging conditions, namely during infection.

Minor comments:

. L212 "Apiotrichum species classified as saprotrophic (e.g. *A. gracile* or *A. porosum*) showed a similar adaptation (Wilcoxon test $P = 1$) in both pathways and were indistinguishable from other pathogens" What seems to distinguish them is that they do not have a higher Slipid/Carbohydrate ratio than their saprotrophic relatives. This does not seem consistent with a "significantly higher translational efficiency for carbohydrate-related metabolic genes in saprotrophic species". It is not very clear to me what is special about these species and how they contribute to the hypothesis tested here.

Response: The *Apiotrichum* genus represent a very intriguing case where a few pathogenic species seem to have emerged among exclusively saprotrophic species. Our hypothesis is that the *Apiotrichum* species are emerging human pathogens and all the species analysed in this study (with genomic data) are potential human pathogens, undetected or misidentified so far in clinical settings. To our knowledge, the pathogenic species in this genus were detected for the first time in clinical settings in 2009 (10.1128/JCM.00460-09) and 2016 (10.1080/00275514.2019.1637645), respectively. This is in contrast with other pathogens like *Trichosporon*, *Cutaneotrichosporon* or *Cryptococcus* which are well known human pathogens for many decades. The genomic signatures are similar among these pathogens and distinct from the saprotrophic counterparts, which are also similar among them. Therefore, we found it very peculiar that a group of saprotrophic species showed no differences from other pathogens. By testing our approach in other fungal lineages in a follow-up study, we aim to predict potential human pathogens based on genomic data.

We modified the text in order to bring forward some of this information and clarify our choice to include these species and the respective hypotheses.

. The experiments presented in this work indicate that there is no strict segregation between saprotrophic and opportunistic pathogen strains according to high temperature growth, this seems in contradiction with the hypothesis that "adaptation to higher temperatures may play a role in ecological niche expansion and in pathogenicity" (L259), could you please clarify?

Response: We recognize that this might be confusing and we edited the text to clarify this point. In our experiments, we observed that some isolates of known human pathogen species are unable to grow above 33 °C. While high-temperature adaptation is important for human pathogens, we hypothesise that it is not the only determinant of pathogenicity but rather it may contribute to ecological niche expansion and the ability of some species to infect human hosts. Likewise, the ability of certain saprotrophic species to tolerate high temperatures may indicate independent adaptations that are not necessarily linked to pathogenicity.

As examples, we see that both the pathogenic *Cutaneotrichosporon dermatis* and the saprotrophic *Cutaneotrichosporon oleaginosus* grow only up to 33 °C; however, they show distinct codon optimization for the metabolic pathways. On another hand, *Cutaneotrichosporon dermatis* and *Trichosporon asahii* are both pathogenic fungi with similar codon optimization; however, the latter is able to grow at 37 °C.

We edited the text for clarity and the sentence now reads:

“We speculate that adaptation to higher temperatures may play a role in ecological niche expansion and ability to infect human hosts, however, it is likely one of multiple factors involved in pathogenicity.”

. Is growth on lipid-rich medium affecting the expression of small secreted proteins (effectors)?

Response: We did not specifically analyse the expression of proteins or effectors, as it was out of the scope of our study, so we cannot directly address this point based on our current data. However, given that effectors are often linked with host interactions and environmental adaptation, it would be interesting to investigate whether lipid availability influences their expression.

. L226 instead of "adapt", prefer "accomodate" to avoid confusion with the evolutionary term

Response: We replaced the word “adapt” with “adjust”.

. Title suggest the topic is about all kind of genomic signatures of adaptation across all lineages of fungi, whereas this is about one particular signature in one lineage of fungi. This does not diminish the relevance and interest of the manuscript but should be made more explicit from the title.

Response: We acknowledge the concern and clarify that our study investigates multiple aspects of genomic adaptation in two phylogenetically different fungal orders (Trichosporonales and Tremellales). Specifically, we study signatures of adaptation on genome structure (e.g. GC content, genome size), functional subsets of gene content and composition (e.g. tRNAs, CAZymes, genes in metabolic pathways) and also codon usage bias and translation efficiency across major functional genomic pathways (e.g. energy production, cytoskeleton or chromatin structure pathways).

We now include additional analyses where we demonstrate that the observed signatures are driven by lifestyle rather than by phylogenetic relatedness. We also note that signatures of pathogenicity can vary across fungal groups. For example, in some fungal lineages, pathogenicity is linked with genome contraction (e.g. *Malassezia*), while in others with expansion of genes associated with secondary metabolite production (e.g. *Aspergillus*) in comparison to non-pathogenic closely related species.

While a comprehensive analysis of all fungal lineages is beyond the scope of our study, we focus on well-studied groups to establish a foundation for broader investigations. Given the species diversity and the multiple genomic signatures analysed in our study, we believe that the current title accurately reflects our study. Furthermore, we consider that readers of Nature Communication are from diverse fields of research and most may not be familiar with the specific taxa studied. Therefore, we prefer to keep the title as it is and hope the reviewer can follow our arguments.

. Fig 2: the search for genomic signatures distinguishing saprotrophic and opportunistic pathogens might be better addressed by a PCA?

Response: We now provide PERMANOVA analyses, where we decompose the phylogenetic relationships into principal components and test the effects of the phylogenetic structure and lifestyle on the variation on different datasets. These are now available in the Supplementary Table 2.

. Fig 3: Is Wilcoxon test the most adapted for comparing ratios (which might not follow a Normal distribution)?

Response: The Wilcoxon test is a non-parametric test that is suitable for distributions which are Normal or not. One advantage is that it is also less sensitive to outliers than other statistical tests (e.g. t-test). Additionally, the results from Wilcoxon test are congruent with PERMANOVA tests, which does not make assumptions on distribution or sample homogeneity. Therefore, we believe that these results are reliable from the statistical point of view.

. A N-terminal translational ramp is typical of codon-optimized genes (10.1016/j.cell.2010.03.031). Is this ramp detected in lipid and carbohydrate-related genes? more or less so between saprotrophic and pathogenic species?

Response: This is indeed an interesting aspect of codon optimization and might be an interesting question for future analyses. However, in our study we focus on the role of the overall codon usage bias on pathway level. We believe that this provides a comprehensive view on the translational adaptation given how complex these pathways are.

. signal peptides are biased towards translationally inefficient codons (10.1016/j.bbamem.2018.09.010) . Was this taken into account when comparins lipid and carbohydrate related genes?

Response: We acknowledge that signal peptides can influence codon bias, however since our analyses consider the entire coding sequence, we believe that it captures the overall trends in codon usage, and any bias is present in every genome, since all genomes were processed and analysed in the same way. Additionally, we analysed genes involved in transport and metabolism, which also include intracellular proteins. Therefore, we did not consider signal peptides in our analyses, besides when addressing secreted proteins.

. Fig 4: The slope of the correlation between S and lag time is steeper for growth on carbohydrates-rich medium than on lipid-rich medium, how can this be interpreted?

Response: We interpret the slopes based on the protein diversity in each pathway. In fungi, the carbohydrate metabolic pathway contains functionally redundant proteins and multifunctional proteins, while in the lipid pathway, the proteins involved are rather specific for their function. The codon optimization of genes involved in carbohydrate metabolism might influence several subprocesses in the pathway. In addition to this, the metabolism of carbohydrates releases quicker and directly available energy. In contrast, the metabolism of lipids involves additional steps, such as the breakdown of lipids into fatty acids and subsequent β -oxidation, which delays energy availability and could be reflected on the slope of these correlations.

We added the following sentences to the Discussion:

“Our experimental data link codon optimization in genes related to certain metabolic pathways to increased fitness in the respective substrates. We propose that codon optimization and improved translation efficiency represent a mechanism of faster adaptation (Fig. 4). Physiological adjustment time (here determined as the lag-time), was more strongly reduced in carbohydrate-rich conditions than in lipid-rich conditions, which may be influenced by the codon optimization in the corresponding metabolic genes. In fungi, the carbohydrate metabolism involves redundant and multifunctional proteins, while lipid metabolism relies mainly on functionally specific proteins. We speculate that the codon optimization in genes involved in carbohydrate metabolism may involve a multitude of subprocesses. In addition, the energy released during carbohydrate metabolism is more immediately

available than for lipid metabolism, which requires additional processing. Hence, we propose that the difference in translation efficiency for carbohydrate and lipid metabolism may reflect a difference in complexity of the underlying cellular processes.”

Reviewer #4 (Remarks to the Author):

Response: We appreciate the time and effort in reviewing our codes and data. Our code and data in the GitHub repository were updated to reflect the modifications on our analyses, including the new analyses.

Point-to-point response to reviewers' comments

Reviewer #1 (Remarks to the Author):

The authors have addressed all the points raised in the original review with text edits or supplemental analysis.

The manuscript explores the hypothesis that the transition of saprotrophic to pathogenicity is facilitated by adaptive translation of relevant metabolic pathways associated with each lifestyle. This work builds on a growing field that recognizes adaptive evolution on translational efficiency as important for trait evolution. It faces several of the same challenges as other works in the field including limitations in genomic tRNA annotation. Nevertheless it contributes to our understanding of the evolution of pathogenicity.

Reviewer #1 (Remarks on code availability):

Code appears sufficient for re-analysis.

Response: We appreciate the very constructive comments, as well as the recognition of our efforts and the value of our study.

Reviewer #2 (Remarks to the Author):

Short summary

In their revised manuscript, Guerreiro et al. provide a more detailed analysis of phylogenetic signal across genome-wide metrics. They conclude that while phylogenetic relatedness explains variation in most metrics, lifestyle (pathogen vs. saprotroph) partly explains variation in codon optimisation of lipid metabolism-related genes (Table S2). They also analyse codon optimisation (S values) in single-copy orthogroups and find no overall difference between pathogens and saprotrophs. However, they report higher relative codon optimisation of metabolism-related genes in pathogens (Figure S19).

Short assessment

The additional analyses improved the manuscript. However, as presented, the results on codon optimisation are misleading and main conclusions remain speculative.

Response: We appreciate all the constructive comments and all the suggestions to improve the clarity and accuracy of the text.

Main points

1. The codon optimisation analysis is misleading. The text suggests direct comparisons of S values (which are largely similar between lifestyles), but the main results (and conclusions) rely on S ratios, whose biological interpretation regarding translational efficiency is unclear.

Response: We agree that our conclusions rely on comparisons of S ratios rather than absolute S values, and we acknowledge the importance of clearly distinguishing between the two. We revised the text in the Results and Discussion sections to refer to S ratios when appropriate and clarified that these represent relative codon optimization between specific metabolic pathways or comparisons of S for specific metabolic pathways and the genome-wide average.

While the biological interpretation of S ratios may be less direct than the interpretation of absolute S values, we use these ratios to compare codon usage among pathways. A higher codon optimization in a given pathway, relative to another pathway or to the genomic background, may reflect an optimization of translation and thereby a footprint of natural selection. We revised the text to make this interpretation clearer.

We added the following sentences to the “Translation efficiency of metabolic pathways predicts fungal lifestyle” subsection of the Results:

“To account for interspecific variation in codon usage bias, S values were normalized by comparing the mean S values across genes associated with lipid and carbohydrate metabolic pathways (S ratio) or by normalizing pathway-specific S values to the genome-wide mean S (S normalized) (see Materials and Methods). A higher relative codon optimization was interpreted as a proxy improved translation efficiency.”

We added the following sentences to the “Codon optimization and tRNA adaptation index estimation” subsection of the Materials and Methods:

“In addition, the S value was normalized (S norm) for each pathway in relation to the genomic S value, to allow direct comparison between pathways among species. A higher codon optimization in a given pathway, relative to another pathway or to the genomic background, may reflect functional prioritization during translation.”

2. The reported R² values (Pearson? Spearman?) do not account for phylogenetic non-independence (e.g. via phylogenetic independent contrasts).

Response: We now provide the correlation values calculated using phylogenetically independent contrasts (PICs). We also clarify which model (Spearman) was used to test for correlations. The Supplementary Figures 3 and 15 were updated to consider phylogenetic independent contrasts. These new results are consistent with our previous analyses and add further support to our final conclusion. The values mentioned in the text were updated.

3. While S is an established proxy for translational selection, the use of "translation" throughout is confusing, particularly without experimental validation. "Codon optimisation" is more precise and appropriate for the audience.

Response: We revised the text and replaced “translation” with “codon optimization” where appropriate.

In text

line 199-209 : this part reads more related to the previous paragraph titled "tRNA composition and expansion are independent of lifestyle"

Response: This paragraph was moved to the previous section.

line 219 : typo "within in these pathways"

Response: We corrected this.

line 221-222 : "common opportunistic pathogenic species well-defined lifestyles" reads confusing.

Response: We rephrased the sentence. It now reads:

“Furthermore, the genera *Cryptococcus*, *Trichosporon*, and *Cutaneotrichosporon* contain species with clearly defined and experimentally supported lifestyles, either as exclusive saprotrophs (e.g. *Cryptococcus amyloletus*) or as common and well-characterized opportunistic pathogens (e.g. *Cryptococcus deneoformans*)³³.”

line 221-222 : misleading, looking at figure S16 I see that ratios are being compared, not S values. It should be clear in the text. In addition the list of genes assigned to each functional category as presented in the Table S9 is missing.

Response: We rephrased the sentence to clarify the use of ratios. It now reads:

“First, we normalized the codon optimization (S) values against the genome-wide S for all genes and compared the normalized S across pathways involved in metabolism (...)”

Additionally, we now provide a table (Supplementary Table 14) with the list of all genes predicted in this study, functional assignments and the respective predicted tAI values.

line 230-233 : misleading as here again ratios are being compared, not S values (although they are presented in Figure S17).

Response: We clarified this and now provide the P values for both S and normalized S (ratio against the genome-wide S). It now reads:

“Additionally, we found that the absolute S and the normalized S values were similar for genes involved in both lipid and carbohydrate metabolic pathways (Wilcoxon test $P = 0.77$ and $P = 0.80$, respectively) in opportunistic pathogenic species (Supplementary Fig. 17).”

line 236 : presented statistics are for ratios and not S values.

Response: We clarified this. It now reads:

“The direct comparison of single-copy orthologous genes detected in all species further suggested higher normalized S values (Wilcoxon test $P = 0.046$) in genes (...)”

The concept of normalized S values is introduced in the previous paragraph.

line 238 : not clear in Figure S19 what the ratio is (S lipid + S carbohydrate / S genome ?)

Response: We clarified this. The figure legend now reads:

“Direct comparison of the codon optimization (S values) among single-copy orthogroups. Protein sequences of genes involved in carbohydrate and lipid transport and metabolism were clustered into orthogroups. The S value was determined based on single-copy orthogroups present in all tested species with well-defined lifestyles (*Cryptococcus*, *Cutaneotrichosporon*, and *Trichosporon*). The S ratio represents the relative codon optimization between both pathways (S lipid : S carbohydrate) for each species.”

line 242-244 : reads vague, plus S are not predictive of lifestyle but S ratios.

Response: We rephrased this. It now reads:

“Lifestyles were distinguishable based on the relative codon optimization (S ratios) of metabolic pathways, which may be relevant to survival in the respective habitats (Fig. 3 and Supplementary Table 2)”. ”

line 252-254 : confusing, it reads like P relates to the comparison of both pathways.

Response: We moved the P value for clarity. It now reads:

“Additionally, *Apiotrichum* species classified as saprotrophic (e.g. *A. gracile* or *A. porosum*) showed a similar adaptation for both pathways and were also indistinguishable from other pathogens (Wilcoxon test $P = 1$, Fig. 3 and Supplementary Fig. 18).”

line 281-286 : clarify here that correlations are for metabolism-specific S values (and not genome-wide) in link with the growth media. Therefore "results suggest that a higher codon optimisation of metabolic pathways .."

Response: We clarified this. It now reads:

“These results suggest that a higher codon optimization of metabolic pathways may provide a fitness advantage by allowing faster adaptation to the surrounding environment and quicker onset of growth.”

line 309 : "is linked to a higher relative codon optimisation of the lipid metabolic pathway". tRNA gene pool not linked to lifestyle according to paragraph "tRNA composition and expansion are independent of lifestyle" and translation not experimentally tested.

Response: We rephrased this. It now reads:

“Our results suggest that the emergence of pathogenicity may be associated with codon optimization for the lipid metabolism. We further show that some species with an optimized codon composition for lipid metabolism, can grow at higher temperatures. Taken together, our findings reflect adaptations relevant for infection of mammal hosts.”

line 356-357 : "the role of codon optimisation in translational speed"?

Response: We rephrased this. It now reads:

“For example, in human cells, the translation rate of genes with optimal codon usage is 58% faster (4.9 codons/s) than non-optimized genes (3.1 codons/s)⁴⁴, emphasizing the role of codon optimization in translational speed.”

line 371 and abstract : "ecology" does not appear in results.

Response: We removed the mention of ecology in the Abstract and Discussion.

line 372-373 : the sentence reads redundant and confusing

Response: We rephrased this. It now reads:

“We detected expansion of the translation machinery (i.e. tRNAs) independently of lifestyle (Supplementary Results), while the translation efficiency of metabolic genes showed signatures of adaptation (i.e. codon optimization) linked to lifestyle.”

Reviewer #3 (Remarks to the Author):

As previously mentioned, this manuscript puts forward very interesting hypotheses on fungal pathogen evolution taking an original angle. The authors have made a commendable effort at clarifying the text and refining some analyses. I appreciated that the authors considered the use of one or multiple isolate per species (comment from reviewer 1) and tested for phylogenetic signal in several analyses (comments from reviewer 2). However, efforts at providing further experimental support has been limited. I found that the included revisions are not yet completely satisfactory and would need to be strengthened by additional information.

Response: We appreciate the feedback and the recognition of our revisions. We have further revised the text to clarify the mentioned issues.

Important points:

1. According to the introduction, “the primary aim of [this] study was to identify genomic signatures and physiological traits associated with lifestyle and host/substrate specialization that could be used to identify potential human fungal pathogens”. A key finding here is that lag phase on carbohydrates and lipids is anticorrelated with translation efficiency of the corresponding genes (Fig. 4). The extent to which this finding enables the identification of potential human pathogens remains unclear. Is there evidence for substrate specialization in the selected species and how it relates to lifestyle needs to be clarified. More thorough comparative analysis of lipid and carbohydrate metabolic pathways would be useful to determine whether orthogroups showing dramatic translation efficiency variation across species are ecologically relevant.

Response: We agree that the relevance of the anticorrelation between phenotype and codon optimization was not clear in the text. We now clarify this point in the Discussion.

We added the following sentences to the Discussion:

“Substrate specialization is ecologically relevant in the context of the transition from saprotrophic to pathogenic lifestyles. The opportunistic pathogen *Cryptococcus deneoformans* shows an increased expression of lipid-related genes during infection^{39,40}. This further suggests that in mammal hosts, lipid-rich environments may select for opportunistic pathogens with optimized lipid metabolism.”

“These results suggest that codon optimization patterns could serve as potential predictive genomic markers for fungal lifestyles.”

Regarding the orthogroups, we previously analysed single-copy orthogroups involved in lipid and carbohydrate metabolism (Supplementary Figure 19). This analysis revealed that the codon optimization (S values) for the two pathways was similar in opportunistic pathogens, whereas saprotrophic species displayed a significantly higher codon optimization for genes involved in carbohydrate metabolism. This supports the idea that lipid metabolism is selectively optimized in pathogens, reflecting an adaptation to lipid-rich environments, such as mammal hosts. This also suggests that the ability to degrade carbohydrates and lipids is equally important for pathogenic species, which colonize both natural environments and mammal hosts.

For species associated with mammals, lipid metabolism is a vital trait. For example, *Cryptococcus neoformans* is a saprotrophic yeast but also an opportunistic human pathogen. This opportunistic pathogen shows increased expression of lipid-related gene during murine infection (Fan et al. 2005, Hu et al. 2008).

In our study, we also show that pathogenic *Cryptococcus* and other pathogenic species exhibit a higher codon optimization for genes involved in lipid metabolism compared to saprotrophic species, including saprotrophic *Cryptococcus* (Supplementary Figure 16, Supplementary Figure 17, Supplementary Figure 19). This supports the hypothesis of an evolutionary adaptation to mammals by opportunistic pathogens. Furthermore, our results also indicate that opportunistic pathogens show similar levels of codon optimization in both lipid and carbohydrate metabolic genes, while saprotrophic species show significantly higher codon optimization for carbohydrate-related genes (Supplementary Figure 17). This suggests an adaptation of opportunistic pathogens to both mammal and natural environments, whereas saprotrophic species are better adapted to natural environments.

Our results emphasize the importance of the lipid metabolism in opportunistic pathogens and carbohydrate metabolism for both saprotrophs and opportunistic pathogens. The patterns observed in the codon optimization analyses are further supported by our experimental data, which, as pointed out by the reviewer, statistically support a link between the codon optimization and the phenotype on lipid or carbohydrate substrates.

Overall, our study links codon optimization with the respective ecological context. While further experimental validation is needed, our results suggest that relative codon optimization of lipid metabolic genes may serve as a genomic signature for pathogenicity.

Fan, Weihua; Kraus, Peter R.; Boily, Marie-Josée; Heitman, Joseph (2005): *Cryptococcus neoformans* gene expression during murine macrophage infection. In *Eukaryotic cell* 4 (8), pp. 1420–1433. DOI: 10.1128/EC.4.8.1420-1433.2005.

Hu, Guanggan; Cheng, Po-Yan; Sham, Anita; Perfect, John R.; Kronstad, James W. (2008): Metabolic adaptation in *Cryptococcus neoformans* during early murine pulmonary infection. In *Molecular microbiology* 69 (6), pp. 1456–1475. DOI: 10.1111/j.1365-2958.2008.06374.x.

2. section "Translation efficiency of metabolic pathways predicts fungal lifestyle". what is the predictive value of the S ratio? In other words, what is the likelihood of correctly predicting lifestyle when shuffling S values for key metabolic pathways between species (null model)?

Response: To test for the predictive value of the S ratio (S lipids : S carbohydrates), we performed a permutation test and a decision tree model test. For this, we used subset of species with well characterized lifestyles (i.e. *Cryptococcus*, *Trichosporon* and *Cutaneotrichosporon*).

For the permutation test, we performed 9999 iterations which resulted on a significant empirical p-value ($P = 0.008$), indicating that the S ratio is able to significantly explain lifestyle.

Additionally, we used a C5.0 boosted decision tree model with rule-based output to test the predictive accuracy of the S ratio (S lipids : S carbohydrates). First, we built a model based on all 18 isolates, which was able to classify the isolates correctly without errors (100% accuracy). In the next approach, from 18 genomes, we randomly sampled 14 genomes (4 saprotrophs and 10 pathogens) to train a model and used the remaining 4 genomes (1 saprotroph and 3 pathogens) to test the model. We performed 9999 iterations as well. In this approach, the model was able to predict all pathogenic species (100% accuracy). For the saprotrophic species, the accuracy was of 80%, due to one isolate (*Cryptococcus amyloletus*) being consistently misclassified (100% inaccuracy). The remaining saprotrophic species were correctly classified (100% accuracy).

The inaccuracy of predicting *C. amyloletus* lifestyle is due to the low sample size in the training dataset and to the S ratio in this species being on the upper limit of the thresholds characterizing a saprotrophic lifestyle, defined by the algorithm during training.

We are confident of the predicting accuracy of the S ratio, for the studied species. We are currently working on testing this model on a wider range of species for a follow-up publication.

We added this paragraph to the Results section:

“In order to test the predictive value of the S ratio (S lipid : S carbohydrates), we performed a permutation test and trained a decision tree classification model³⁴ using the *Cryptococcus*, *Trichosporon* and *Cutaneotrichosporon* species. Based on the permutation test, the S ratio was able to significantly explain lifestyle (empirical P value = 0.016). Additionally, the decision tree model trained on the full dataset classified the lifestyle accurately for all isolates (100% accuracy). To further test model robustness, we randomly sampled four saprotrophs and ten pathogens to train a model and used the remaining one saprotroph and three pathogens to test the model. In this cross-validation approach, all pathogenic species were correctly predicted (100% accuracy). For the saprotrophic species, the accuracy of the model was of 80%, due to one species (*Cryptococcus amyloletus*) being consistently misclassified due to the S ratio being near the classification threshold (Supplementary Table 11). The remaining saprotrophic species were correctly classified (100% accuracy). Using this decision tree approach, we conclude that the S ratio is a suitable predictor of fungal lifestyle in the studied species, particularly for distinguishing opportunistic pathogens from saprotrophic species.”

We added the respective information to the Materials and Methods.

3. In spite of their arguments, I think the authors need to come up with a more specific and accurate title for their study (see also Reviewer 1 comment 4). First, the number of genes encoding CAZymes and other secreted is sometimes used as a signature for fungal lifestyles, whereas this parameter does not appear discriminating in the present study. Second, the lack of specificity suggests that genomic signatures have been assessed systematically, including for instance any type of gene family, although this study is focused on testing specific hypotheses. Third, the authors agree in their rebuttal letter that further functional studies will be required to establish a causal link between codon usage and fitness, but this causal relationship is implicit in the current title.

Response: In our study, we compare several genomic traits (e.g., CAZyme content, GC content) that have previously reported to be associated with lifestyle in other fungal groups. While CAZymes and most other traits did not distinguish lifestyles, we identified codon optimization as a genomic feature strongly associated with lifestyle. We applied robust comparative and statistical methods to identify such associations. We consider the expression “genomic signatures” appropriate, as it reflects the identification of genomic features that correlate with lifestyle. Importantly, we interpret “signatures” as consistent genomic patterns associated with a lifestyle, rather than causality. We also believe that the current title is accessible and informative for the broad readership of Nature Communications. If the editor believes that the title must be change, we are happy to do so.

Suggestion: the authors compared codon optimization (S) across several pathways. A more systematic approach would be to search for pathways enriched among genes with the highest codon optimization. Has this approach been tested?

Response: Instead of analysing pathway enrichment among genes with the highest codon optimization values, we compared S values between lifestyles across all annotated COG functional categories (Supplementary Figure 16). This allowed us to compare full pathways and identify consistent differences in codon optimization between lifestyles. While enrichment analyses can be useful, they rely on thresholds (e.g. genes with the 5% highest S values) and may miss broader trends across entire functional groups. We believe that our approach is suitable to detect genomic signatures associated with lifestyle across the genome, without introducing biases.

Minor comments and suggestions:

. L250 the S ratio used in figure 3 should be introduced in the main text

Response: We now introduce the S ratio in the main text. We added the following sentences:

“To account for interspecific variation in codon usage bias, S values were normalized by comparing the mean S values across genes associated with lipid and carbohydrate metabolic pathways (S ratio) or by normalizing pathway-specific S values to the genome-wide mean S (S normalized) (see Materials and Methods). A higher relative codon optimization was interpreted as a proxy improved translation efficiency.”

. Fig4 "Data are presented for isolates with consistent growth patterns across four independent replicates" > need to represent variability of the data along the Y axis

Response: We updated the figure to depict the variability.

. The section "Saprotrophic species grow at mammal body temperatures" is not related to codon usage analysis and does add predictive power to the approach. how many times was the experiment repeated (this is not mentioned in sup fig. 21). still appears disconnected to the rest of the manuscript (reviewer 2 minor comment 4)

Response: This experiment was performed for 4 replicates per isolate. We have now added this information to the figure legend as well the variation among replicates. To provide further support to this figure, we now include qualitative growth data (Supplementary Figure 22). For this figure, we repeated the same growth experiment at different temperatures, but in 5 mL of medium in reaction tubes. We provide the photos of each tube at each condition. The results are similar between the different approaches.

We agree that this section is not directly related to codon optimization. However, it complements our genomic analyses by assessing a critical virulence factor, such as the ability to grow at mammal body temperatures. We show that some saprotrophic species (e.g., *Vanrija* spp.) exhibit codon optimization patterns similar to those of opportunistic pathogens. Additionally, we also show that these species can also grow at mammal body temperatures. These results support the hypothesis that they may possess thermotolerance and translational adaptations relevant for pathogenicity in a mammal host.

We added the following paragraph to this section of the Results to clarify the meaning of these results:

“Some saprotrophic species (e.g. *Vanrija* spp.) showed similar codon optimization patterns to those of opportunistic human pathogens (Figure 3 and Supplementary Fig. 20). The growth temperature experiments (Figure 5, Supplementary Fig. 21 and Supplementary Fig. 22) complement these findings by showing that some saprotrophic species are able to grow at mammal body temperature (e.g. *Vanrija* spp.), similar to opportunistic human pathogens. Together, these results suggest that some saprotrophic species may have the thermo- and translational adaptation associated with pathogenicity, further suggesting that these may emerge as potential human opportunistic pathogens.”

. "host infection arose independently multiple times": based on the ancestral state reconstruction provided, that opportunistic pathogenicity arose 3 times or that it was lost 3 times in the lineages analyses seems equally likely to me

Response: We agree that both gain and loss of pathogenicity are possible scenarios. However, we consider that the gain of pathogenicity is a more likely scenario. This is suggested by the distribution of pathogenic species across at least three phylogenetically distinct clades, that are separated by saprotrophic lineages and by that most fungal species in the Tremellales are not mammal pathogens. These patterns suggest independent adaptations to mammal hosts rather than a single ancestral pathogenic state followed by multiple losses.

. L186: "we observe that the tRNA composition was similar among distantly related species (Supplementary Fig. 9), suggesting convergent evolution with respect to tRNA genes." In this figure, the similarity in tRNA composition decreases with phylogenetic distance at the genus level but not beyond – this does not seem sufficient to infer some form of convergent evolution

Response: We agree that the similarity in tRNA composition between genera alone is insufficient to infer convergent evolution. However, we also detect a low phylogenetic signal (Blomberg's $K < 1$) for most tRNA genes (38 out of 45 genes), indicating that closely related species are less similar in tRNA composition than expected under a Brownian motion model of evolution. These results suggest that tRNA gene composition may be under selective pressure independent of phylogeny, which is consistent with convergent evolution. However, in the current study, we are not able to identify the specific drivers involved in this.

. L188: “the most frequently used synonymous codons did not consistently correspond with the most abundant tRNA gene families in the genomes” – this seems to argue against adaptive translation at whole genome level, is there a better match between codon usage and tRNA repertoire when considering specific gene subsets?

Response: In our analyses, we see evidence for adaptive translation on pathways relevant for the lifestyle (Supplementary Figure 16). The codon optimization value (S) integrates codon usage, effective number of codons, tRNA genes copy number and GC content (on the 3rd codon position). Therefore, the codon usage and tRNA repertoire is integrated in the S value. By directly comparing the relative synonymous codon usage (RSCU) and the tRNA repertoire (Supplementary Table 8), we are able to identify differences between the RSCU of the whole genome and the subset of genes involved in lipid or carbohydrate metabolism. However, given the complexity of the codon usage and tRNA pool data, we consider that the comparison of S values is more informative and reliable than the direct comparison between only RSCU and tRNA repertoires.

Reviewer #3 (Remarks on code availability):

I have not tested the code but appreciate that the authors make it available. The readme file might be expanded to help readers navigate among the different files provided.

Response: We appreciate the suggestion and we updated the readme with additional details.

Reviewer #4 (Remarks to the Author):

Response: We appreciate all the constructive comments and we are glad that our manuscript could contribute to the training of a Researcher.

Point-to-point response to reviewers' comments

Reviewer #2 (Remarks to the Author):

In this revised version, the authors have incorporated a phylogenetic perspective into their correlation analyses, although the methods used for this part are not described.

Response: We now provide the respective details in the Materials and Methods.

They have also clarified the results concerning codon optimization. However, the main conclusions still rely on comparing ratios of codon optimization values across different functional categories, and for which the biological meaning remains unclear. Since S values are directly comparable across species, there is no need to "account for interspecific variation" as stated in the text and methods. In their analysis, S values do not differ between pathogens and saprotrophs, contradicting the statement that codon optimization predicts lifestyle.

Response: We have now added a paragraph in the Discussion (please see below) to address the biological meaning of the different S values (absolute, normalized and ratio).

We are happy to provide additional clarifications; however, it is unclear which results are being referred to by the reviewer. All of our analyses consistently show that the S values are able to distinguish between pathogens and saprotrophs. The only exceptions are within the 1) *Apiotrichum* genus, which we discuss that the species classified as saprotrophic might be emerging pathogens (e.g. Figure 3); and 2) in the single-copy orthogroups comparison (Supplementary Figure 19), where the patterns are still observable although the p-values are significant but close to 0.05 due to the reduced size and power of the single-gene dataset.

Reviewer #3 (Remarks to the Author):

The authors have addressed previous comments adequately.

I still think the title of the manuscript lacks specificity and clarity but I will let the Journal Editor decide on that matter.

One point that may deserve further discussion: the authors argue that gene content is not a determinant of lifestyle in these fungal orders (L162) and later quickly elude the role of transcription (L412) with no justification. A short discussion on how the regulation of gene expression at the transcription level (in addition to protein translation) may determine lifestyle transitions, and the possible molecular mechanisms involved, seems like an important element of context to provide.

Response: We added the following paragraph to the Discussion to address these concerns:

“Gene expression and regulation are directly affected by several molecular mechanisms, including chromatin remodelling, epigenetic modifications, transcription factor binding, and the use of alternative promoters^{45,46}. Transcriptional reprogramming can be linked to shifts between saprotrophic and host-associated lifestyles, for example via the upregulation of stress-response pathways, secreted enzymes, or virulence factors^{47,48}. Hence, transcription regulation defines the repertoire of expressed genes, while translational adaptation regulates the protein synthesis efficiency to meet specific requirements. These mechanisms suggest that ecological transitions are enabled by both transcriptional regulation and translational adaptation. While we show that translational adaptation provides an advantage in niche adaptation (Figure 4), future studies addressing the relevance of transcriptional regulation and how this interacts with translation efficiency will be crucial to fully understand the molecular mechanisms involved in lifestyle transitions.”

Overall, the revised manuscript provides convincing evidence for a role of adaptive translation in transition to pathogenic lifestyle in a fungal lineage, and clearly discusses the impact and limitations of the study. This is an insightful and original contribution to our understanding of fungal evolution.

Response: We appreciate the recognition of our study and all the constructive feedback throughout the reviewing process.

Please see below my opinion on concerns on NCOMMS-24-59407B regarding the use of S ratio:

S values measure the degree of co-adaptation between the tRNA pool of a cell and codon usage in genes. They take into account both individual gene sequences (codon usage) and the cellular background (tRNA pool, generally inferred from number of copies in the genome).

S values are directly comparable across species (as said by reviewer 2) in the sense that if S is higher for gene1 in species X than in Y, it means the co-adaptation between tRNAs and codon usage for gene1 is better in X than in Y.

But a strictly identical gene sequence (same codon usage) will have different S values across species depending on the cellular background (tRNA pool). This means that the distribution range of S values in a genome varies from species to species. In other words, with a same S value, gene1 can be the best optimized gene in species X and just average relative to other genes in species Y. Its degree of co-adaptation with tRNAs will be identical strictly speaking, but the meaning for the cell functioning will be different if many other genes outcompete it for tRNAs recruitment (species Y) or not (species X). This is probably what the authors call “interspecific variation”.

Response: This is a correct interpretation of the S values between different species.

For this reason, I think the ratio approach of the authors has merits and offers biological insights by showing with a single metric which of two pathways has better co-evolved with the tRNA pool

across species. But I agree with reviewer 2 that they need to be more specific so that their conclusions are not confusing. Typically, if conclusions are based on S1/S2 ratio comparisons, then the "relative codon optimization of pathway 1 and 2 predicts lifestyle" and not simply "codon optimization".

Response: We agree with the suggestion and we revised the text to further emphasize that our results and conclusions are based on the relative codon optimization instead of codon optimization.

In short, my recommendations would be as follows:

1) - More clearly present and comment S values for each pathway in each species, as well as the distribution of S values in all the genomes analyzed (as supplementary material) to document the effect of codon usage and tRNA pool variation on the S metrics, so that the choice of ratios is clearly justified

Response: We provide now an additional panel to Supplementary Figure 17 (Supplementary Figure 17B), where we show the distribution of the genome-wide, absolute and normalized S values. In this analysis, the saprotrophic species clearly show a higher codon optimization (S) for the genes involved in carbohydrate metabolism in comparison to lipids. The S value in opportunistic pathogens for both pathways is much closer to each other, suggesting co-optimization of the codon usage. We also observe variability in the genome-wide S value among the species.

The ratio between both pathways and the normalization of each pathways against the genome-wide value, allow for comparison across species, considering the genomic background of each individual.

B
2) - Another approach to compare individual genes across species would be to quantile normalize S values across genomes. Individual normalized S values would then be directly comparable across species (top S values will be equal in all species). Personally, I had favored normalization as it makes the whole S value distributions comparable and does not rely on one reference gene/pathway as the ratio does. I would suggest the authors to try this approach and verify that they reach the same conclusions as with ratios.

Response: We appreciate the suggestion of quantile normalization.

We would like to emphasize that the S value is a single value that represents the co-adaptation between the tRNA gene pool and the codon usage bias of a given dataset, and further accounts for the effective number of codons and GC content at the 3rd position.

In our analyses, we normalized the pathway S values against the genome-wide S (Supplementary Figures 16 and 17). In these analyses, we also compared these normalized values with the absolute values. Both metrics (Supplementary Figures 16 and 17), as well as the S ratios (Figure 3 and Supplementary Figure 18), provided consistent results. In addition, we also compared the S value of single copy orthologous genes present in all genomes analysed by this approach (Supplementary Figure 19). This approach provided further support to the previous findings based on the ratio, although the statistical signal was weaker ($P = 0.046$) due to low sample size.

Our current normalization approach accounts for the genomic background of codon usage and tRNA pool composition, allowing direct comparisons across species without relying on S ratios. Quantile normalization would equalize the distributions by rank and thereby force all genomes to have identical S distributions, removing the biological information regarding codon optimization intensity. By considering the genome-wide S as a reference, we can infer whether a metabolic pathway is more strongly optimized than expected based on the genomic background. We therefore consider the genome-wide normalization to be more appropriate to address our research questions, as it considers the genomic context, while allowing robust comparisons across species.

3) - Invite authors to carefully revise their manuscript to make sure interpretations and conclusions are strictly accurate, that the analysis strategy is properly and fully justified. A short paragraph dedicated to the interpretation of the S ratio/normalized S/raw S values and their meaning in the context of this study might be useful to add to the discussion section.

Response: We added the following paragraph to the Discussion to clarify the interpretation of the different values:

“In this study, we used different representations of codon optimization (S) to capture complementary aspects of translational adaptation. Absolute S values directly reflect the co-adaptation between codon usage bias and the genomic tRNA pool within each genome, while also integrating the effective number of codons and the GC content at the third position. Since individuals and species often show variations in the tRNA pool and codon usage biases, absolute S values may not be directly comparable across genomes. To account for different genomic backgrounds, we calculated normalized S values, which express the codon optimization of a given pathway relative to the genome-wide S value ($S_{\text{absolute}} : S_{\text{genome}}$). This metric indicates whether a pathway is more or less optimized than expected within the same genome. Finally, S ratios provide a direct comparison between two functional categories within a genome ($S_{\text{pathway 1}} : S_{\text{pathway 2}}$), which reflects whether a pathway is more strongly optimized relative to another. Altogether, we consider the three metrics to provide complementary insights: absolute S values assess the global level of translational adaptation in a genome, normalized S values emphasize deviations from the genomic baseline, and S ratios determine relative optimization between pathways. By combining these metrics, we assess the contribution of codon optimization to fungal lifestyle adaptation.”

Reviewer #3 (Remarks on code availability):

I have not tested the code but appreciate that the authors make it available

Response: We have updated the code to reflect the revisions.